# Learning the Inverse Temperature of Ising Models under Hard Constraints

**Rohan Chauhan**
University of California, Irvine
rmchauha@uci.edu

**Ioannis Panageas**
University of California, Irvine
ipanagea@uci.edu

## Abstract

We consider the problem of estimating the inverse temperature parameter $\beta$ of an $n$-dimensional truncated Ising model using a single sample. Given a graph $G = (V, E)$ with $n$ vertices, a truncated Ising model is a probability distribution over the $n$-dimensional hypercube $\{-1, 1\}^n$ where each configuration $\sigma$ is constrained to lie in a truncation set $S \subseteq \{-1, 1\}^n$ and has probability $\Pr(\sigma) \propto \exp(\beta\sigma^\top A\sigma)$ with $A$ being the adjacency matrix of $G$. We adopt the recent setting of [Galanis et al. SODA'24], where the truncation set $S$ can be expressed as the set of satisfying assignments of a $k$-SAT formula. Given a single sample $\sigma$ from a truncated Ising model, with inverse parameter $\beta^*$, underlying graph $G$ of bounded degree $\Delta$ and $S$ being expressed as the set of satisfying assignments of a $k$-SAT formula, we design in nearly $\mathcal{O}(n)$ time an estimator $\hat{\beta}$ that achieves a consistency rate of $\mathcal{O}(\Delta^3/\sqrt{n})$ with the true parameter $\beta^*$ for $k \gtrsim \log(d^2k)\Delta^3$.

Our estimator is based on the maximization of the pseudolikelihood, a notion that has received extensive analysis for various probabilistic models without [Chatterjee, Annals of Statistics '07] or with truncation [Galanis et al. SODA '24]. Our approach generalizes recent techniques from [Daskalakis et al. STOC '19, Galanis et al. SODA '24], to confront the more challenging setting of the truncated Ising model.

## 1 Introduction

Markov random fields (MRFs) are a common framework for analyzing high-dimensional distributions with complex conditional structures. A well-studied example of an MRF and the primary topic of inquiry in this paper is the *Ising model* (Ising, 1925), a probability measure $\mu_{G,\beta}$, over all assignments $\boldsymbol{\sigma}$ in the binary hypercube $\{-1, 1\}^n$. The model is parameterized by a graph $G = (V, E)$ and inverse temperature $\beta$, taking the form $\mu_{G,\beta} \propto \exp\left(\beta\boldsymbol{\sigma}^\top A\boldsymbol{\sigma}\right)$, where $A$ being the adjacency matrix of $G$. The simplicity of the Ising model has led to widespread adoption in fields as disparate as statistical physics, finance, the social sciences, and computer vision, among others (see (Chatterjee, 2007; Qin & Zhao, 2011; Harris, 2013; Wang et al., 2015) and the references therein for a brief collection of examples). These applications have, in turn, motivated a substantial body of research on efficient sampling (Bresler, 2015; Lubetzky & Sly, 2013; Sly & Sun, 2012), rigorous testing (Daskalakis et al., 2019a), and principled inference of the inverse temperature parameter and interaction matrix (Dagan et al., 2020; 2021) under the framework of the Ising model.

Beginning from the work of (Chatterjee, 2007), there has been substantial interest in the task of estimating inverse temperature parameter $\beta$, given only one sample $\boldsymbol{\sigma} \sim \mu_{G,\beta}$ and the graph $G$, using the maximum pseudolikelihood estimator of (Besag, 1975). This setup subsumes the setting of multi-sample estimation as $\ell$ samples of an Ising model over $n$ nodes is equivalently a single sample of an $n\ell$ Ising model over $\ell$ disconnected components of a graph. This line of work is driven by the inherent technical constraints of network data, wherein it is often impractical to obtain independent observations of the same network responses (Daskalakis et al., 2019b; Dagan et al., 2020). Interestingly, despite both static and dynamic phase transitions in model behavior as $\beta$ varies which can render sampling computationally intractable (NP-hard) (Galanis et al., 2016), it remains possible to construct consistent estimators of $\beta$ and the interaction matrix $A$, provided that $\beta = \mathcal{O}(1)$ (Dagan et al., 2020; 2021; Daskalakis et al., 2019b; Mukherjee et al., 2022a).

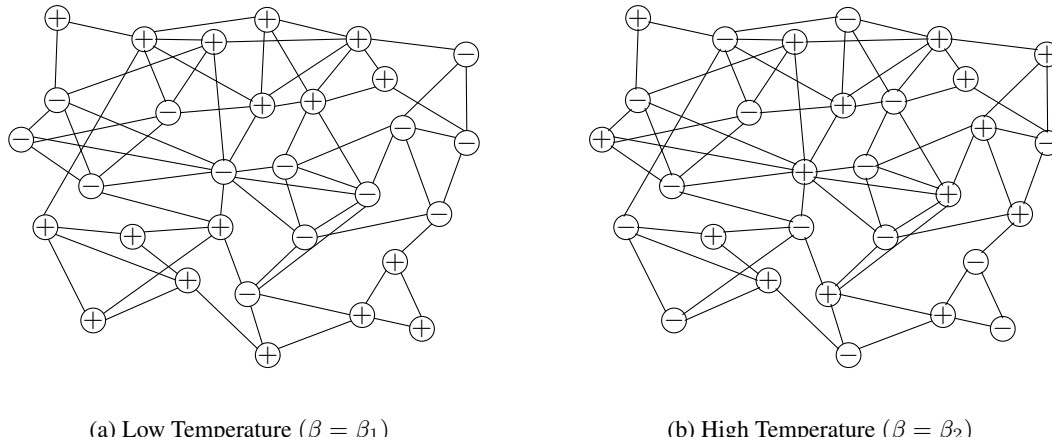

(a) Low Temperature ($\beta = \beta_1$)  (b) High Temperature ($\beta = \beta_2$)

Figure 1: Two typical spin configurations over the Ising model at temperatures $\beta_1, \beta_2$ with $\beta_1 \gg \beta_2$ (equivalently at a lower temperature $T_1$ and a higher temperature $T_2$ where $T \propto 1/\beta$). Each node has spin $+1$ or spin $-1$. The left panel shows the configuration at a lower temperature, with alignment producing large domains of positive and negative spin assignments, while the right panel exhibits a more disordered pattern.

In many real-world applications, however, we face not only *soft* constraints, which influence the model behavior by introducing correlations or dependencies, while the full support of the measure remains intact, but also *hard* constraints: Certain configurations are outright forbidden, and entire regions of the configuration space are excluded from the support of the distribution. Such hard-constrained models (also called *truncated*) arise naturally in applications involving high-dimensional, interconnected systems with strict feasibility requirements. One particularly notable example of which arises in the context of spatial transcriptomics—a biological framework for characterizing gene and protein expression in cells within organic tissue, relative to their spatial organization. The local relationships between cells are often represented as nodes in a graph with edges linking cells together that are close in physical space (Eng et al., 2019). Associated with each node is a set of measurements of expression to capture how the relationships between cells impact the phenotype of a given node. The complex relationship between genes often forces certain configurations of expressions to be *infeasible*. In fact, the phenomenon of *lateral inhibition* can cause a cell expressing gene one to prevent its neighbors from expressing it as well, instead causing them to express gene two, as seen in the Notch-Delta pathway (Ghosh & Tomlin, 2001). The *hard constraints* discussed in the above setting are not unique to spatial transcriptomics and are also commonly found in the context of channel assignments in communication networks (Zafer & Modiano, 2006), carrier-sense multiple access networks (Durvy & Thiran, 2006; Durvy et al., 2009), and multicasting networks (Karvo et al., 2002; Luen et al., 2006) among others.

In this work, we study the problem of parameter estimation in $n$-dimensional Ising models that are hard-constrained to the satisfying assignments of a bounded-degree $k$-SAT formula $\Phi$ expressed in CNF (Conjunctive Normal Form), using one sample. This means that we have access to a sample from an Ising model, conditioned that it only takes values in a subset $S \subset \{\pm 1\}^n$ that is represented through the satisfying assignments of a $k$-SAT formula, adopting the framework from (Galanis et al., 2024). The choice of the $k$-SAT framework is motivated by the fundamental observation that any truncation set $S \subseteq \{\pm 1\}^n$ can be exactly realized as the solution space of such a formula, provided the number of clauses is sufficiently large. This representation is particularly advantageous for statistical inference, as the structural parameters, in particular the clause size $k$ and the variable degree, serve as natural controls for the complexity of the induced correlations in the analysis of the Maximum Pseudolikelihood Estimator (MPLE).

More generally, learning in truncated MRFs using one or multiple samples has been studied in the context of discrete product distributions truncated by the set of satisfying assignments of $k$-SAT formulas (Galanis et al., 2024; 2025), more generally by truncated sets with combinatorial structure (Fotakis et al., 2022), the hard-core model, and integer valued spins constrained over proper $H-$colorings (Blanca et al., 2018; Bhattacharya & Ramanan, 2021). Our key deviation from the

aforementioned works stems from the fact that the Ising model is *not a product distribution*, and common tools used to control the concentration of measure on the hypercube do not apply. This, moreover, induces two sources of interdependence, namely from the model itself and from the structure of the truncation set. With this background in mind, we seek to address the following challenge.

> *Is it possible to efficiently learn discrete distributions with complex dependencies under hard constraints, having access to a single sample?*

## 1.1 Our Results

Our main contribution is an affirmative answer to the previous challenge, by providing a sufficient condition on the $k$-SAT formula that induces the truncation set, in terms of the maximum degree $\Delta$ of $G$. We begin by formally defining the class of truncated Ising measures that is the primary inquiry of this work. Given a graph $G$, with associated adjacency matrix $A$, and inverse temperature $\beta$ we define the pmf of a *truncated* Ising model for any $\boldsymbol{\sigma} \in \{\pm 1\}^n$ to be

$$\mathbf{Pr}_{\beta,S}(\boldsymbol{\sigma}) := \frac{1}{Z_{\beta,S}} \exp\left(\beta\boldsymbol{\sigma}^\top A\boldsymbol{\sigma}\right) \mathbf{1}\{\boldsymbol{\sigma} \in S\}, \qquad \text{(Truncated Ising Model)}$$

where $\mathbf{1}\{\boldsymbol{\sigma} \in S\}$ captures the indicator function determining if $\boldsymbol{\sigma} \in S$ and $Z_{\beta,S}$ is a renormalization term denoted the partition function. In our case, $S$ is expressed as the set of satisfying assignments of a bounded degree $k$-SAT formula $\Phi_{n,k,d} = \Phi$ in constrained normal form. Formally, let $\Phi$ be a formula in conjunctive normal form over variables $\mathcal{V} = \{v_1, \dots, v_n\}$ and clause set $\mathcal{C}$, subject to the structural constraints that each clause contains exactly $k$ literals (variables or their negations) and each variable occurs in at most $d$ clauses. We identify each configuration $\boldsymbol{\sigma} \in \{-1, 1\}^n$ with a truth assignment via the mapping where variable $v_i$ is *true* if $\sigma_i = 1$ and *false* if $\sigma_i = -1$. A configuration $\boldsymbol{\sigma}$ is then said to satisfy $\Phi$ if and only if every clause $C \in \mathcal{C}$ contains at least one literal that evaluates to *true* under this assignment. Recall, any subset of the hypercube can be represented as the set of satisfying assignments of a degree-$d$ $k$-SAT formula, provided that $d$ is sufficiently large.

Our main result – stated below – is a sufficient condition on the degree $d$ of the formula $\Phi$ in terms of the size of each clause $k$ and the maximum degree $\Delta$ of the underlying graph $G$ of the Ising model, for computationally and statistically efficient estimation of the inverse temperature parameter $\beta$.

**Theorem 1** (Informal Version of Theorem 2). *Let $\boldsymbol{\sigma}$ be a single sample from a truncated, $n$-dimensional Ising model with inverse temperature $\beta^*$, where the truncation set is captured by the satisfying assignments of a $k$-SAT formula $\Phi_{n,k,d}$ and the underlying graph $G$ has maximum degree $\Delta$ of order $o(n^{1/6})$. For $n$ sufficiently large, $\beta^*$ is $\mathcal{O}(1)$ and $k \geq \Omega(4\Delta^3(1 + \log(d^2k + 1)))$, there exists an $\mathcal{O}(\Delta^3 n \log(n))-$algorithm which takes as input $\boldsymbol{\sigma}$ and outputs an estimator $\hat{\beta}$ such that*

$$\mathbf{Pr}_{\beta^*,S}\left[|\hat{\beta} - \beta^*| \leq \frac{c\Delta^3}{\sqrt{n}}\right] \geq 99\%, \ \text{for a constant } c > 0 \text{ independent of } n, \Delta, d, k.$$

*Remark* (Consistency). Notice when $\Delta$ is $\mathcal{O}(1)$, our estimate achieves $\mathcal{O}(1/\sqrt{n})$-consistency, matching the minimax rate for parameter estimation. Likewise, the restriction of $\Delta$ to be on the order of $o(n^{1/6})$ ensures the error of our estimator $\hat{\beta}$ is asymptotically diminishing and in turn consistent.

*Remark* (Lower Bounds). We further note that Galanis et al. (Galanis et al., 2024) recently established a lower bound for learning discrete Boolean distributions, demonstrating that estimation is information-theoretically impossible when $k \leq \log(d) - \log(k) + \Theta(1)$. This result relies on constructing a $k$-SAT instance with a single satisfying assignment. Importantly, this lower bound extends to our setting, implying that our results are optimal up to a constant factor when $G$ is a graph of bounded degree. While subsequent work (Galanis et al., 2025) has provided a tighter bound, those techniques do not immediately translate to our context; the quadratic Hamiltonian induces a rugged energy landscape distinct from boolean product setting, necessitating different analytical tools.

## 1.2 Technical Overview

Given a single-parameter exponential family like we focus on, a natural approach to estimating the parameter is to find the maximum likelihood estimate. However, the computational intractability of

the partition function $Z_{\beta,S}$ for Ising models (see (Galanis et al., 2016) and the references therein) renders this approach infeasible. In light of these challenges, we utilize the maximum pseudolikelihood estimator introduced by (Besag, 1975) and provided below.

$$\hat{\beta} := \arg\max_{\tilde{\beta}} \prod_{i \in [n]} \mathbf{Pr}_{\tilde{\beta},S}(\sigma_i|\boldsymbol{\sigma}_{-i}) = \arg\min_{\tilde{\beta}} -\sum_{i \in [n]} \log(\mathbf{Pr}_{\tilde{\beta},S}(\sigma_i|\boldsymbol{\sigma}_{-i})) := \arg\min_{\tilde{\beta}} \phi(\tilde{\beta};A,\boldsymbol{\sigma}).$$

(MPLE)

We note that the second equality holds due to the monotonicity of the $\log(\cdot)$ function. Towards demonstrating the consistency of the maximum (log)-pseudolikelihood estimate $\hat{\beta}$, we follow the first and second derivative paradigm outlined by Chatterjee (Chatterjee, 2007; Daskalakis et al., 2019b; Galanis et al., 2024; 2025), which involves showing,

- $\mathbf{Pr}_{\beta^*,S}[\nabla_\beta\phi(\beta^*;A,\boldsymbol{\sigma}) \leq \mathcal{O}(\sqrt{n})] \geq 1 - o(1)$,
- $\inf_{\beta \in (-B,B)} \nabla_\beta^2\phi(\beta;A,\boldsymbol{\sigma}) \geq \Omega(n/\Delta^3)$ with probability $1 - o(1)$ over $\boldsymbol{\sigma} \sim \mathbf{Pr}_{\beta^*,S}$.

The first condition ensures the derivative of the log-pseudolikelihood objective with respect to the true model parameters $\beta$ divided by $n$ is *close* to $0$, which is the value of the gradient of $\phi$ computed at the estimator, which in turn implies $\beta$ is an *approximate stationary point of the objective*. Moreover, by demonstrating that the second derivative of the objective $\nabla_\beta^2\phi(\beta;A,\boldsymbol{\sigma})$ is $\Omega(n/\Delta^3)$-strongly convex with probability $1 - o(1)$ over a draw of the truncated Ising model, it implies that approximate stationary points of the objective are close in Euclidean distance to the optimum. These two facts combine to show the proximity of the optimum of the log-pseudolikelihood objective to $\beta$.

Showing both of these conditions hold simultaneously is made complex due to the highly non-uniform measure induced by conditional dependencies of both the interaction matrix $A$ and the truncation set $S$. To demonstrate the first condition, we craft upper bounds on the variance of the first derivative of $\phi$, using the technique of exchangeable pairs pioneered by (Chatterjee, 2007), which, when combined with Chebyshev's inequality, implies an upper bound in probability. The primary challenge of this work lies in establishing the second condition. Previous works which used the deterministic structure of the interaction matrix to guarantee the concavity of the objective. By contrast, in our setting, the second derivative is governed by the local geometry of the truncation set $S$ in the vicinity of the sample $\boldsymbol{\sigma}$. The Hessian $\nabla_{\tilde{\beta}}^2$ is computed by summing a function of the local fields $m_i(\boldsymbol{\sigma}) = \langle A_i, \boldsymbol{\sigma}\rangle$, where the support of this sum is limited to the valid neighbors of $\boldsymbol{\sigma}$ in $S$ with respect to the Hamming distance [1]

To show that, with high probability under the truncated Ising model, a sample $\boldsymbol{\sigma}$ has many neighboring configurations at Hamming distance 1, we construct an argument based on the Lovász Local Lemma (LLL), to guarantee the existence of a large number of satisfying assignments to $\Phi$ that differ from $\boldsymbol{\sigma}$ in exactly one bit. Using this powerful tool, however, requires control of the probabilities of partial spin assignments, which, given the tendency of the Ising model to contract into arbitrarily small portions of the hypercube and exhibit long-range correlations, can prove challenging. Counteracting this, our argument conditions on nodes outside of a specially crafted independent set $I$ of the graph $G$, which preserves the marginal distribution of any given spin $\mathbf{Pr}_{\beta^*}[\sigma_i|(\sigma_1,...,\sigma_{i-1},\sigma_{i+1},...,\sigma_n)]$ despite limited to a small fraction of the support of the measure, and collapses the Ising model into a product measure. To conclude the strong convexity of $\phi(\cdot;A,\boldsymbol{\sigma})$ it remains to show a lower bound on $m_i(\boldsymbol{\sigma})$ in probability via a coupling argument which exploits the underlying edge structure of the connectivity graph $G$.

We additionally note that the recent results on estimating Ising models using the pseudolikelihood approach (Dagan et al., 2020) rely on sophisticated concentration inequalities derived from the fast mixing nature of Glauber dynamics on the Boolean hypercube and their relation to the Gibbs measure in order to conclude the strong convexity of the PL objective and bound the gradient with high probability. In our model, these powerful tools are not applicable due to the fragmented nature of the truncation set, rendering the domain of our measure into disconnected islands and make the Glauber dynamics *non-Ergodic*; the inequalities only imply concentration within a connected component of $S$, which may be too small to be informative. We lastly emphasize that our polynomial time algorithm *does not* scale with the size of the truncation set, enabling inference despite $S$ being small with regards to the entire hypercube.

---

[1] The Hamming distance metric counts the number of differing indices between two vectors. Thus, a neighbor at Hamming distance one is obtained by flipping exactly one index of $\boldsymbol{\sigma}$.

## 1.3 RELATED WORK

The literature of parameter estimation in Markov Random Fields, and over hard-constrained and truncated measures, is vast. In light of this, we mention a brief collection of works relevant to our setting, and defer additional background and discussion to the appendix. Single sample estimation initiated by (Besag, 1975; Chatterjee, 2007) has yielded a rich bounty of results ranging from the setting of the Ising model (Chatterjee, 2007; Bhattacharya & Mukherjee, 2018; Ghosal & Mukherjee, 2020; Dagan et al., 2020), peer dependent logistic regression (Daskalakis et al., 2019b; Mukherjee et al., 2022a; Daskalakis et al., 2020), higher order Ising models (Mukherjee et al., 2022b), and robust inference over discrete distributions (Diakonikolas et al., 2021). (Bhattacharyya et al., 2021) demonstrated the feasibility of single-sample learning in the context of the hard-core model, a size-weighted distribution over all independent sets in a graph $G$; following up on this, (Galanis et al., 2024; 2025) studied parameter inference in a product distribution truncated by the satisfying assignments of a $k$-SAT formula. The hard-constrained models studied in this work are a subset of the literature analyzing efficient parameter estimation and learning in truncated (Daskalakis et al., 2019c; 2018; Fotakis et al., 2022; De et al., 2023; Nagarajan & Panageas, 2020) and censored distributions (Lugosi et al., 2024; Plevrakis, 2021; Fotakis et al., 2021).

## 2 PRELIMINARIES

### 2.1 NOTATION

We denote the set of $\{1, 2, ..., n\}$ as $[n]$. Vectors $\mathbf{x} \in \mathbb{R}^d$ are denoted with boldface, and matrices $M \in \mathbb{R}^{m \times n}$ with capital letters. Given a vector $\mathbf{a} = (a_1, a_2, \ldots, a_n)$ and a subset $I \subseteq [n]$, let $\mathbf{a}_I$ denote the length-$|I|$ coordinate vector $\{a_i : i \in I\}$, and $\mathbf{a}_{-i}$ denote the vector $\mathbf{a}$ with the $i-$th element removed. We denote the probability of an event $\mathcal{A}$ over the *untruncated* measure parameterized by $\beta$ as $\mathbf{Pr}_{\mu_\beta}$ and over the *truncated* Ising measure as $\mu_{G,\beta,S}(\mathcal{A}) = \mathbf{Pr}_{\beta,S}[\mathcal{A}] = \mathbf{Pr}_{\mu_\beta}[\mathcal{A}|S]$. We often remove the explicit dependence on $S$ and $G$ for clarity of explanation.

We will say an estimator $\hat{\beta}$ is consistent with a rate $\mathcal{O}(f(n))$ (or equivalently $f(n)$-consistent) with respect to the true parameter $\beta^*$ if there exists an integer $n_0$ and a constant $C > 0$ such that for every $n \geq n_0$, with probability at least $1 - o(1)$,

$$|\hat{\beta} - \beta^*| \leq Cf(n).$$

Lastly, we call an entry $\sigma_i$ of $\boldsymbol{\sigma}$ to be *flippable* if both $(\sigma_i, \boldsymbol{\sigma}_{-i})$ and $(-\sigma_i, \boldsymbol{\sigma}_{-i})$ lie in $S$, and moreover we denote by $e_i(\boldsymbol{\sigma})$ the indicator of the event that $\sigma_i$ of $\boldsymbol{\sigma}$ is flippable.

### 2.2 MAXIMUM PSEUDO-LIKELIHOOD ESTIMATION

Towards explicitly computing the (log)-pseudolikelihood objective and its associated derivatives, we begin by finding the *conditional* distributions of the individual spins conditioned on the rest of the assignment, $\mathbf{Pr}_\beta(\sigma_i|\boldsymbol{\sigma}_{-i})$. Notice, when $\sigma_i$ is *not* flippable, the conditional distribution is trivially one, while for flippable $i$, the probability is given by the following:

$$\mathbf{Pr}_\beta(\sigma_i|\boldsymbol{\sigma}_{-i}) = \frac{\exp(\beta m_i(\boldsymbol{\sigma})\sigma_i)}{\exp(-\beta m_i(\boldsymbol{\sigma})) + \exp(\beta m_i(\boldsymbol{\sigma}))} \text{ where } m_i(\boldsymbol{\sigma}) := \sum_{j=1}^n A_{ij}\sigma_j.$$

Denoting $\mathcal{F}(\boldsymbol{\sigma})$ to be the set of flippable variables in $\boldsymbol{\sigma}$, the negative log pseudo-likelihood objective $\phi(\beta; A, \boldsymbol{\sigma})$ can be written explicitly as follows:

$$\begin{aligned}
\phi(\beta; A, \boldsymbol{\sigma}) &:= -\sum_{i \in \mathcal{F}(\boldsymbol{\sigma})} \log\left(\mathbf{Pr}_\beta(\sigma_i|\boldsymbol{\sigma}_{-i})\right) \\
&= \sum_{i \in \mathcal{F}(\boldsymbol{\sigma})} \left(\log\left(\exp(-\beta m_i(\boldsymbol{\sigma})) + \exp(\beta m_i(\boldsymbol{\sigma}))\right) - \beta m_i(\boldsymbol{\sigma})\sigma_i\right).
\end{aligned} \tag{2.1}$$

In the sequel, we drop the reference to $A$ in the pseudo-likelihood when the interaction matrix is clear. The first and second derivatives of the objective (2.1) with respect to the inverse temperature

parameter $\beta$, denoted by $\phi_1(\beta; \boldsymbol{\sigma}) = \nabla_\beta \phi(\beta; A, \boldsymbol{\sigma})$, $\phi_2(\beta; \boldsymbol{\sigma}) = \nabla_\beta^2 \phi(\beta; A, \boldsymbol{\sigma})$ are given below:

$$\phi_1(\beta; \boldsymbol{\sigma}) = \sum_{i \in \mathcal{F}(\boldsymbol{\sigma})} (m_i(\boldsymbol{\sigma})(\tanh(\beta m_i(\boldsymbol{\sigma})) - \sigma_i)), \quad \phi_2(\beta; \boldsymbol{\sigma}) = \sum_{i \in \mathcal{F}(\boldsymbol{\sigma})} \frac{m_i(\boldsymbol{\sigma})^2}{\cosh^2(\beta m_i(\boldsymbol{\sigma}))}.$$

Note that the negative log-pseudo-likelihood is convex as the second derivative is *always* non-negative (sum of squares).

## 2.3 An Auxiliary Lemma

In our analysis, demonstrating the consistency of our estimator, we frequently require the existence of combinatorial objects, such as satisfying assignments to a $k$-SAT formula or fulfilling specific constraints without their explicit construction. Such existence questions often reduce to avoiding a collection of undesirable events, each of which occurs with low probability and exhibits limited dependence on the others. This setting is naturally addressed using the probabilistic method, and in particular, the Lovász Local Lemma.

**Lemma 2.1** (Symmetric Lovász Local Lemma). *Given a collection of events $\{A_i\}_{i \in [n]}$, where each event $A_i$ satisfies $\Pr(A_i) \leq p$ and each event is mutually independent from all but at most $d$ other events. If*

$$e \cdot p \cdot (d+1) \leq 1, \tag{Symmetric LLL}$$

*then $\mathbf{Pr}\left(\bigcap_{i=1}^n \overline{A_i}\right) > 0$, where $\overline{A_i}$ denotes the complement of $A_i$ and $e$ refers to Euler's number.*

## 3 Learning Truncated Ising Models

In this section, we prove our main result, i.e., we provide a sufficient condition on the "complexity" of the $k$-SAT formula $\Phi$, (and by extension the truncation set $S$) in terms of the size of the clauses $k$, the degree of the formula $d$ and the maximum degree of the graph $\Delta$ for efficient estimation of the inverse temperature parameter $\beta$. In advance of proving our result, we lay out some mild assumptions on the interaction matrix $A$ and Ising model $\mu_{G,\beta,S}$ which have been employed in past works (Galanis et al., 2024; 2025; Dagan et al., 2020; Daskalakis et al., 2019b; Chatterjee, 2007; Bhattacharya & Mukherjee, 2018).

**Assumption 1.** *Within our model (Truncated Ising Model), we assume*

- *$A$ is the adjacency matrix of a connected graph over $n$ nodes, with maximum degree $\Delta$ being $o(n^{1/6})$ and entries $A_{ij} \in \left\{-\frac{1}{\Delta}, +\frac{1}{\Delta}\right\}$ representing positive or negative interactions.*

- *The inverse temperature parameter $\beta$ lies in the open interval $(-B, B)$.*

- *The truncation set $S$ is the set of satisfying assignments to a $k$-SAT formula $\Phi$ in conjunctive normal form.*

*Remark* (Assumptions). The assumption of graph connectivity ensures that the interaction matrix contains sufficient signal energy for consistent parameter estimation. This requirement plays a role analogous to the lower bounds on the Frobenius norm of the interaction matrix posited in (Chatterjee, 2007; Daskalakis et al., 2019b; Bhattacharya & Mukherjee, 2018). Absent such a structural guarantee, the log-partition function $Z_{\beta,S}$ may fail to diverge asymptotically with $n$, which prevents the pseudolikelihood objective from achieving consistency. This pathology is exemplified by the Curie-Weiss model with couplings scaling as $1/n$; there, the Frobenius norm of the interaction matrix is $\mathcal{O}(1)$. In contrast, our connectivity and degree assumptions imply that the squared Frobenius norm of $A$ scales as $\Omega(n/\Delta^2)$, which diverges to infinity, thereby ensuring the problem is well-posed.

While general specifications of the Ising model allow for arbitrary coupling strengths $A_{ij} \in \mathbb{R}$, we restrict the interaction magnitudes to a uniform value $|A_{ij}| = 1/\Delta$ towards isolating the structural component of peer influence. This constraint posits that the social pressure exerted by any single neighbor is functionally equivalent, thereby modeling a "peer effect" (Bertrand et al., 2000; Sacerdote, 2001; Duflo & Saez, 2003) where individual decisions are driven by the collective consensus of the local group rather than the idiosyncratic intensity of specific dyadic relationships. By imposing this homogeneity, we ensure that the model captures the emergent coordination arising from network topology and group alignment, rather than being dominated by outliers with arbitrarily strong pairwise connections.

The formal version of our main result is given as follows.

**Theorem 2** (Main result). *Let $\boldsymbol{\sigma}$ be a single sample from a truncated, $n$-dimensional Ising model satisfying Assumption 1. For all $k \geq \frac{4\Delta^3(1+\log(d^2k+1))}{\log(1+\exp(-2B))}$, if $n$ is sufficiently large, there exists an $\mathcal{O}(\Delta^3 n \log n)-$time algorithm which takes as input $\boldsymbol{\sigma}$ and outputs an estimator $\hat{\beta}$ such that*

$$\mathbf{Pr}_{\beta^*, S}\left[|\hat{\beta} - \beta^*| \leq \frac{c\Delta^3}{\sqrt{n}}\right] \geq 99\%, \text{ for a constant } c > 0 \text{ independent of } n, \Delta, d, k.$$

*Remark* (The algorithm). We compute $\hat{\beta}$ in time $\mathcal{O}(\Delta^3 n \log n)$ by running projected gradient descent (PGD) on the normalized log-pseudolikelihood objective $n^{-1}\phi(\beta; \boldsymbol{\sigma})$. Standard results from convex optimization (e.g., (Boyd & Vandenberghe, 2004)) imply that PGD converges to an $\epsilon$-optimal solution in $\mathcal{O}(\kappa \log(1/\epsilon))$ iterations, where $\kappa$ is the condition number of the objective, that is the ratio of the smoothness (i.e., the Lipschitz constant of the gradient) to the strong convexity parameter. In the sequel (Section 3.3), we demonstrate that the pseudo-likelihood objective is strongly convex with parameter $\Omega(n/\Delta^3)$, implying the normalized objective is $\Omega(1/\Delta^3)$-strongly convex. This, combined with an upper bound of 1 on the norm of the normalized gradient of the log-pseudolikelihood, yields the condition number of the objective is $\kappa = \mathcal{O}(\Delta^3)$. Due to the statistical limitations of the pseudolikelihood estimator, whose distance from the true parameter $\beta$ can be as large as $\mathcal{O}(\Delta^3/\sqrt{n})$, we set $\epsilon = 1/\sqrt{n}$. Obtaining an accuracy of $\epsilon$, requires $\mathcal{O}(\Delta^3 \log n)$ iterations, each requiring $\mathcal{O}(n)$ time, resulting in an overall runtime of $\mathcal{O}(\Delta^3 n \log n)$.

To prove Theorem 2, we begin by explicitly demonstrating how the conditions on the first and second derivatives of the $\phi(\beta; \boldsymbol{\sigma})$ imply the consistency of the MPLE $\hat{\beta}$ in Section 3.1. We then establish the conditions on the first derivative in Section 3.2, and the second in Section 3.3.

## 3.1 ROADMAP FOR PROVING THEOREM 2

In this subsection, we demonstrate the relationship between the derivatives of the (log)-pseudolikelihood and the estimation error $|\hat{\beta} - \beta^*|$.

**Lemma 3.1.** *Let $\beta^* \in (-B, B)$ be the true parameter of the truncated Ising model $\mu_{G, \beta^*, S}$ and $\hat{\beta}$ be the MPLE. It follows that with probability $1 - o(1)$*

$$|\hat{\beta} - \beta^*| \leq \frac{|\phi_1(\beta^*; \boldsymbol{\sigma})|}{\min_{\tilde{\beta}} \phi_2(\tilde{\beta}; \boldsymbol{\sigma})}$$

*Proof Sketch.* We relate $\hat{\beta}$ with $\beta^*$, via smooth interpolation of both the parameter values themselves $\beta(t) = t\hat{\beta} + (1-t)\beta^*$, and the gradient $s(t) = (\hat{\beta} - \beta^*)\phi_1(\beta(t); \boldsymbol{\sigma})$. As the derivative of the pseudolikelihood at $\hat{\beta}$ is zero, we note that $s(1) = 0$. The fundamental theorem of calculus implies

$$-(\hat{\beta} - \beta^*)\phi_1(\beta^*; \boldsymbol{\sigma}) = s(1) - s(0) = \int_0^1 s'(t)dt = (\hat{\beta} - \beta^*)^2 \int_0^1 \phi_2(\beta(t); \boldsymbol{\sigma})dt.$$

The lemma follows from $\int_0^1 \phi_2(\beta(t); \boldsymbol{\sigma})dt \geq \min_{\tilde{\beta} \in (-B, B)} \phi_2(\tilde{\beta}; \boldsymbol{\sigma})$ and $\phi_2(\tilde{\beta}; \boldsymbol{\sigma}) \geq 0$. □

With this lemma in hand, demonstrating Theorem 2 reduces to showing $\phi_1(\beta^*; \boldsymbol{\sigma}) = \mathcal{O}(\sqrt{n})$ and $\phi_2(\beta; \boldsymbol{\sigma}) = \Omega(n/\Delta^3)$ simultaneously with probability $1 - o(1)$.

## 3.2 ANALYSIS OF FIRST MOMENT

The lemma below establishes the upper bound on $\phi_1(\beta; \boldsymbol{\sigma})$. To demonstrate an upper bound on $\phi_1(\beta; \boldsymbol{\sigma})$ in probability, we use the technique of exchangeable pairs (Chatterjee, 2007) to construct a bound on its variance. With the variance controlled, we invoke Markov's inequality to conclude $\phi_1(\beta; \boldsymbol{\sigma})$ that concentrates around its mean.

**Lemma 3.2** (Upper Bound on $\phi_1(\beta; \boldsymbol{\sigma})$ in Probability). *Fix a constant $\delta > 0$. The log-pseudolikelihood $\phi(\beta; \boldsymbol{\sigma})$ of a truncated Ising model fulfilling Assumption 1 satisfies the following upper bound in probability, for all $\beta \in \mathbb{R}$*

$$\mathbf{Pr}_\beta\left[|\phi_1(\beta; \boldsymbol{\sigma})| \leq \sqrt{\frac{(12 + 4B)n}{\delta}}\right] \geq 1 - \delta.$$

## 3.3 Second Derivative Bound

For reference, we recall the expression for the Hessian of the log pseudo-likelihood,

$$\phi_2(\beta; \boldsymbol{\sigma}) = \frac{\partial^2 \phi(\beta; \boldsymbol{\sigma})}{\partial \beta^2} = \sum_{i=1}^{n} \frac{m_i^2(\boldsymbol{\sigma})}{\cosh^2(\beta m_i(\boldsymbol{\sigma}))} e_i(\boldsymbol{\sigma}).$$

The primary aim of this section is to demonstrate the following lower bound in probability.

**Lemma 3.3** (Lower Bound on $\phi_2(\beta; \boldsymbol{\sigma})$ in Probability). *The log-pseudolikelihood $\phi(\beta; \boldsymbol{\sigma})$ of a Ising model, truncated by a $k$-SAT formula with $k \geq \frac{4\Delta^3(1+\log(d^2 k+1))}{\log(1+\exp(-2B))}$, fulfilling Assumption 1 satisfies the following lower bound in for all $\beta \in (-B, B)$*

$$\mathbf{Pr}_{\beta^*}\left[ \frac{\partial \phi^2(\beta; \boldsymbol{\sigma})}{\partial^2 \beta} \geq \frac{n \exp(-B)}{\Delta^3(8kd)^2} \right] \geq 1 - \frac{(24 + 8B)}{n^{0.1}}.$$

We prove this claim in two steps, by **firstly** guaranteeing there are a *linear* number of flippable variables $v_i \in V$, which contribute to the value of the second derivative, and **secondly** ensuring the value of each term in the sum is bounded below by a constant.

### 3.3.1 Ensuring Flippability

Given a sample $\boldsymbol{\sigma}$, the flippability of a variable $\sigma_i$ under the $k$-SAT formula $\Phi$ is characterized by the condition that every clause containing $\sigma_i$ is satisfied by at least one other variable in the clause. Consequently, $\sigma_i$ is *not* flippable if there exists a clause $C$ such that all other variables $v_j \in C\backslash\{v_i\}$ are assigned values that fail to satisfy the clause—an *antagonistic* configuration. Under our assumptions, the truncated Ising model may be defined at arbitrarily low temperatures, including values of $\beta = \mathcal{O}(1)$ that exceed the critical threshold. In this regime, standard concentration-of-measure tools, such as log-Sobolev inequalities or Dobrushin-type conditions, are no longer valid and fail to yield meaningful bounds. This makes it significantly more difficult to lower bound the probability of antagonistic configurations, and, by extension, to bound the probability that a given variable is flippable.

Towards providing such a bound, we construct an independent set $I$ within the graph $G$, such that the marginal distribution of the spins within $I$, conditioned on the variables outside of the independent set $V \setminus I$, collapses into a product distribution, circumventing the above difficulties. Indeed, the distribution of $\boldsymbol{\sigma}_I$ conditional on an assignment of the remaining nodes $\boldsymbol{\sigma}_{V \setminus I}$ is given as follows, with $m_i^{V \setminus I}(\boldsymbol{\sigma})$ instead of being random variables, they are now fixed constants.

$$\mathbf{Pr}_\beta(\boldsymbol{\sigma}_I | \boldsymbol{\sigma}_{V \setminus I}) \propto \exp\left( 2\beta \sum_{i \in I} m_i^{V \setminus I}(\boldsymbol{\sigma})\sigma_i \right), \text{ where } m_i^{V \setminus I}(\boldsymbol{\sigma}) = \sum_{j \in V \setminus I} A_{ij}\sigma_j.$$

One of the issues that arises from conditioning our graphical model on $V \setminus I$ is the natural truncation of the $k$-CNF formula $\Phi$; erasing the variables outside of the independent set $I$ from $\Phi$ transforms it into a new formula $\Phi'$, which contains only variables from $I$. An inherent concern in the selection of the independent set is the presence of clauses in $\Phi'$ containing only a few variables, i.e., of size $o(k)$, which can significantly skew the marginal distributions away from uniformity. To address this, we show that there exists an independent set $I \subset V$ that intersects a linear fraction of the variables in *every* clause, ensuring sufficient coverage and mitigating this issue.

**Lemma 3.4.** *Let $G$ be a graph with maximum degree $\Delta$ of order $o(n^{1/6})$ and $\Phi$ be a $k$-SAT formula. If $k > 10\Delta^3(1 + \log(dk\Delta^2))$, then there exists an independent set $I \subset V$ such that $\Phi'$, $\Phi$ truncated on $V \setminus I$, is a $\lambda k$-SAT formula where $\lambda = 1/4\Delta^3$.*

*Proof Sketch.* To begin, we describe an algorithm that maps bijections of the vertex set to independent sets in the graph $G$. Formally, given a map $\rho : V \to [n]$, we construct an independent set by selecting all vertices $u \in V$ such that $\rho(u) > \rho(v)$ for all $v \in N(u)$. This selection criterion ensures that no two adjacent vertices are included in the set, as any edge $\{u, v\} \in E$ prevents both $u$ and $v$ from satisfying the condition simultaneously.

Under the uniform measure over all maps $\rho$, the event that a vertex $v$ is selected into the independent set depends only on the relative rankings under $\rho$ of $v$ and its neighbors. This locality implies that for any pair of vertices $u, v \in V$ with graph distance $d(u, v) \geq 3$, the corresponding selection events are independent. Leveraging this property, for each clause $C$, we can extract a subset $C' \subset C$ consisting of variables whose neighborhoods are pairwise disjoint, implying the event of selection for all elements $v \in C$ are mutually independent, yielding the selection events for all variables in $C'$ are mutually independent. This allows us to treat the number of selected variables in $C' \cap I$ as a sum of independent Bernoulli random variables, enabling the use of Chernoff bounds, and consequently, we obtain an exponential upper bound on the probability of the bad event that $|C \cap I| < \lambda k$.

To establish a bound on $k$ in terms of $d$ and $\Delta$ that ensures the existence of a marking with the desired properties, we invoke the symmetric version of the Lovász Local Lemma (Lemma 2.1). Each variable appears in at most $d$ other clauses, and the bad event corresponding to a variable's inclusion in the independent set depends only on the configuration of variables within its two-hop neighborhood. Since this neighborhood contains at most $\Delta^2 + 1$ variables, each bad event is dependent on at most $kd(\Delta^2 + 1)$ others. By the symmetric Lovász Local Lemma, if $k$ is sufficiently large so that the associated condition is met, then with positive probability, there exists an independent set satisfying the required condition.s

$$2e \exp\left(-\frac{k}{8(\Delta^2 + 1)(\Delta + 1)}\right)(kd(\Delta^2 + 1)) < 1.$$

When $\Delta \geq 5$, if $k \geq 10\Delta^3(1 + \log(dk\Delta^2))$, $k$ satisfies this requirement, completing the proof. $\square$

Armed with the guarantee that the truncated $k$-SAT formula $\Phi'$ contains a sufficient number of variables in each clause, we relate the flippability of a given variable $v_i$ to the satisfiability of select clauses solely through elements of the independent set $I$. Indeed, a sufficient condition for a variable $v_i$ to be flippable is that every clause containing $v_i$ is satisfied by at least one variable in the independent set $I$. We capture this requirement using the following indicator function:

$$s_j(\boldsymbol{\sigma}) := \mathbf{1}\left\{\text{every clause containing } j \text{ is satisfied by some } i \in I\right\}.$$

This reformulation is particularly valuable because it translates the notion of flippability, which originally depends on the full joint distribution of the Ising model at arbitrary inverse temperature $\beta$, into a condition over the structure of the product distribution induced by the independent. As the selection of $I$ can be made independently of the spin configuration and is governed by local rules (e.g., via randomized greedy selection based on random bijections), the probability that $s_j(\boldsymbol{\sigma}) = 1$ can be effectively analyzed using standard concentration inequalities such as Chernoff bounds, enabling explicit probabilistic guarantees on the flippability of variables, despite the tendencies of the underlying Ising model to exhibiting long-range dependencies.

To this end, we now establish a sufficient condition on $k$ that ensures all variables are flippable with constant probability.

**Lemma 3.5.** *Given a sample $\boldsymbol{\sigma} \sim \mathbf{Pr}_{\beta^*, S}$, such that the $k$-SAT formula $\Phi$ which induces the truncation set $S$, satisfies the following clause size bound*

$$k \geq \frac{4\Delta^3(1 + \log(d^2k + 1))}{\log(1 + \exp(-2B))}.$$

[2]*Then for $\Delta \geq 5$, there exists an independent set $I$ following Lemma 3.4 such that*

$$\mathbf{Pr}_{\beta^*, S}[s_j(\boldsymbol{\sigma}) = 1] \geq 1/2 \quad \forall j \in V \setminus I.$$

*Moreover, for any set $V' \subseteq I$ we can find a collection of $R \subseteq V'$ with $|R| \geq |V'|/(2kd)^2$ that are neighborhood disjoint in the interaction graph of $\Phi$ such that for all subcollections $\{i_1, ..., i_t\} \subset R$,*

$$\mathbf{Pr}_{\beta^*, S}\left[e_{i_t}(\boldsymbol{\sigma}) = 1 | e_{i_1}(\boldsymbol{\sigma}) = 1, ..., e_{i_{t-1}}(\boldsymbol{\sigma}) = 1\right] \geq 1/2.$$

---

[2]This term scales as $\gtrsim e^{2B}\Delta^3 \log(d^2k)$.

### 3.3.2 BOUNDING THE MAGNETIZATIONS

It now remains to demonstrate that the squared magnetizations $m_i^2(\boldsymbol{\sigma})$ are bounded below with constant probability over a draw of the truncated Ising model. We begin by providing a conditional lower bound to $m_i(\boldsymbol{\sigma})$.

**Lemma 3.6.** *The magnetizations, $m_i(\boldsymbol{\sigma}) = \sum_{j \in [n]} A_{ij}\sigma_j$, of the truncated Ising model satisfy the following relation.*

$$\mathbb{E}_{\beta^*}[m_i(\boldsymbol{\sigma})^2|\boldsymbol{\sigma}_{-j}] \geq \frac{\exp(-B)}{\Delta^2}\mathbf{Pr}_{\beta^*}[e_j(\boldsymbol{\sigma}) = 1]$$

This lower bound is only non-trivial when *both* realizations $(\sigma_i, \boldsymbol{\sigma}_j)$ and $(\sigma_i, -\boldsymbol{\sigma}_j)$ are feasible under the truncation set, i.e $e_j(\boldsymbol{\sigma}) = 1$; likewise, this term only contributes to the $\phi_2(\beta; \boldsymbol{\sigma})$ if $e_i(\boldsymbol{\sigma}) = 1$. Towards maximizing the second derivative, we wish to select a sequence of edges $(i, j)$ such that both $e_i(\boldsymbol{\sigma}) = 1$ and $e_j(\boldsymbol{\sigma}) = 1$, that is, both endpoints are flippable. As each element $v_i \in I$ has at least one neighbor in $V \setminus I$, and the graph has maximum degree $\Delta$, we can construct a subset $I' \subseteq I$ of size $|I'| > \frac{n}{\Delta^2}$ such that no two elements in $I'$ share any common neighbors. With this independent set $I'$, we define a vertex bijection $h : V \to V$ as follows. For each $v \in I'$, we assign $h(v)$ to be a *unique* neighbor of $v$ in $V \setminus I$. For vertices outside $I'$, we assign the remaining mappings arbitrarily, while maintaining the constraint that $h$ remains a bijection on $V$. Using Lemma 3.6 and the above bijection $h$, we can find a lower bound on the entire conditional second derivative.

**Lemma 3.7.** *Over the truncated Ising model, given a bijection $h : V \to V$ defined by the above procedure, the conditional second derivative satisfies the following first moment bound.*

$$\sum_{i=1}^{n} \mathbb{E}_{\beta^*}[m_i(\boldsymbol{\sigma})^2 e_i(\boldsymbol{\sigma})|\boldsymbol{\sigma}_{-h(i)}] \geq \frac{n\exp(-B)}{2\Delta^3(4kd)^2}.$$

**Establishing Lemma 3.3** Armed with the lower bound on the conditional expectation of $\phi_2(\beta; \boldsymbol{\sigma})$, to obtain our final lower bound we control the variance of $\sum_{i=1}^{n} \mathbb{E}_{\beta^*}[m_i^2(\boldsymbol{\sigma})e_i(\boldsymbol{\sigma})|\boldsymbol{\sigma}_{-h(i)}]$ with the method of exchangeable pairs, in a similar fashion to Lemma 3.2. We then apply Chebyshev's inequality to the conditional variance to obtain our bound in probability.

## CONCLUSION AND FUTURE WORK

In this paper, we present an affirmative answer to the challenge of single sample learning in the truncated Ising model, at all temperatures $\beta \in \mathcal{O}(1)$, giving a sufficient condition for the truncation set $S$ to ensure consistent inference, and extending the existing framework *beyond* boolean product distributions. Towards this goal, we craft concentration inequalities for the first and second derivatives of the log-pseudolikelihood via local connectivity arguments over the domain.

The present work opens the door to important future questions : (i) Given the above framework, does analyzing measures with random Hamiltonians, like those of the Sherrington-Kirkpatrick model, alleviate the dependence on $\Delta$? (ii) Do logistic regression techniques used to estimate untruncated graphical models apply to the constrained setting? (iii) Is there a way to simultaneously remove the $o(n^{1/6})$ assumption on the maximum degree of the graph while improving the rate of consistency?

### ACKNOWLEDGMENTS

Ioannis Panageas is supported by NSF grant CCF-2454115. Part of this work was done while Rohan Chauhan was visiting Archimedes AI.

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

# Contents

## A   Related Work and Additional Background

The Ising model originated as a mathematical model of ferromagnetism on subgraphs of the lattice $\mathbb{Z}^d$, capturing local interactions in physical systems. Ising solved the one-dimensional case in his thesis (Ising, 1925), while Onsager later resolved the two-dimensional case (Onsager, 1944), revealing a continuous phase transition between ferromagnetic and paramagnetic states. Beyond low dimensions, the Ising model also serves as a foundational example of spin glasses, aiding both condensed matter physics and probability theory in understanding complex magnetic materials and high-dimensional correlated loss landscapes (Talagrand, 2003; 2010).

Our inquiry into the Ising model will be statistical in nature, concerning the consistent estimation of the inverse temperature parameter under the presence of truncation using a single sample. Despite the seeming simplicity of this task, the presence of phase transitions yields it to be theoretically impossible in certain regimes; our results stand in light of these challenges. One of the primary difficulties in our task is our graph $G$, by extension interaction tensor $A$, is arbitrary (although of a somewhat bounded degree), and thus our results hold in a regime that is neither fully locally connected or mean-field. The first work to prove such a result was (Chatterjee, 2007), who via use of the technique of exchangeable pairs, a variation on Stein's inequality to prove variance bounds, was able to demonstrate the consistency of the maximum pseudolikelihood estimate derived from a single sample(an objective that will expounded on in the sequel) given the log partition function $F_{G,\beta,n} = \log(Z_{G,\beta,n})$ is diverges with $n$ in the large data limit. As an example, when this seemingly innocuous assumption is not upheld, under the mean field Curie-Weiss model, i.e.

$$\mathbf{Pr}_{CW}(\boldsymbol{\sigma}) = \frac{1}{Z_\beta} \exp\left( \beta \sum_{i,j \in [n]} \frac{1}{n} \sigma_i \sigma_j \right), \qquad \text{(Curie-Weiss)}$$

consistent estimation is *impossible*, as simple calculus yields that $\lim_{n\to\infty} F_{G,\beta,n} = \mathcal{O}(1)$. Moreover, if $\beta$ diverges to infinity with $n$, the psuedolikelihood objective ceases to be strongly concave, collapsing the Fisher information, and rendering estimation impossible. Beyond the estimation of the inverse temperature, (Mukherjee et al., 2022b) was able to extend the regime of Chatterjee (Chatterjee, 2007), demonstrating results for the *joint* estimation of the inverse temperature $\beta$ and the external field $h$. Viewing the task of estimating the inverse temperature as structure estimation over

a parametric class of interaction matrices, i.e parameterized by $\beta$, (Dagan et al., 2020) generalized this setting to provide learning guarantees for large classes of parametric spaces, relying on a clever use of conditioning to use Dobrushin's condition at all constant temperatures.

Beyond single sample learning, viewing the Ising model as a Markov random field, there is a larger body of work devoted to *structure* learning of the graph underlying the model using *multiple* samples. In a breakthrough work Bresler (Bresler, 2015), demonstrated how to effciently estimate the strength of links in the graph underlying the Ising model by way of bounding the marginal influence each node recieves. Building on this, (Hamilton et al., 2017) generalized this work to subsume models with higher order interaction terms and multiple possible spin states.

A parallel line of work has also investigated the feasibility of learning Markov random fields under *hard*-constrained distributions with a *single* sample. This line of work commenced with (Bhattacharya & Ramanan, 2021) studying the fugacity parameter of the hard core model, i.e. a probability distribution over independent sets over a graph $G = (V, E)$ represented by binary vectors $\boldsymbol{\sigma} \in \{0, 1\}^n$, where $\sigma_i = 1$ indicates the node is included in the independent set

$$\mathbf{Pr}_\lambda^N(\boldsymbol{\sigma}) = \frac{1}{Z_{G,\lambda}} \lambda^{\sum_{u=1}^N \sigma_u} \prod_{(u,v) \in E} \mathbf{1}\{\sigma_u + \sigma_v \leq 1\}. \qquad \text{(Hard Core Model)}$$

Following up on this (and closer to our setting), (Galanis et al., 2024) considered the feasibility of learning boolean product distributions over truncated portions of the boolean hypercube, making use of the *tilted-$k$-SAT* model over $S \subset \{0, 1\}^n$, where $S$ is defined to be the set of solutions of a fixed bounded degree $k$-CNF formula $\Phi$.

$$\mathbf{Pr}_\beta(\boldsymbol{\sigma}) = \frac{1}{Z_\beta} \exp\left(\beta \sum_{i \in [n]} \sigma_i\right) \mathbf{1}\{\boldsymbol{\sigma} \in S\} \qquad \text{(Tilted K-SAT)}$$

Lastly, there has been a substantial amount of interest in constrained normal form formulae. The literature is multi-faceted, and we only recount the literature, pertinent to our setting. In the bounded-degree setting, it is well known from (Gebauer et al., 2016) that the satisfiability threshold ( $d \lesssim 2^{k/2}$), that is the regime of the degree parameter with respect to the clause size is guaranteed to have a solution, coincides with the ability to apply the Lovascz Local Lemma (Guo et al., 2019), a powerful application of the probabilistic method. Moreover, there has been substantial inquiry into random $k$-CNF formulae and their solutions, as a function of the clause density $\alpha = m/n$, where $m$ is the number of clauses in the formula and the multitude of other phase transitions governing the intrinsic geometry of the solution space, i.e. how do solutions cluster together and what can be said about local connections between them. Our results in both settings, take hold in the Lovascz Local Lemma regime where a solution under an average draw has many neighbors Hamming-distance one away in $S \subset \{-1, 1\}^n$.

As a first step towards estimating the inverse temperature, our work lays the statistical groundwork to guarantee there exists an objective whose objective yields a consistent estimator for $\beta$ and an algorithm to efficiently find it. This can be seen as a generalization of both regimes, as the Ising model with external field generalizing the tilted-$k$-SAT model by introducing a quadratic interaction term and extending the external field $h$ to be an arbitrary vector in the $n - 1$-sphere, i.e $\|h\|_2 \leq 1$, rather than the all ones vector. Beyond deterministic truncation, our results are the first to hold the broader class of solution sets under random truncation.

## A.1    CONJUNCTIVE NORM FORMULAE

Conjunctive normal form (CNF) is a canonical way of expressing Boolean formulas as a conjunction of disjunctions, or equivalently, as an "AND" of "OR" clauses. Each clause is a disjunction of literals, where a literal is either a Boolean variable or its negation. A CNF formula is said to be a $k$-CNF formula (or a $k$-SAT instance) if every clause contains exactly $k$ literals. For example, $(x_1 \vee \neg x_2 \vee x_3) \wedge (\neg x_1 \vee x_4 \vee x_5)$ is a 3-CNF formula with two clauses.

## A.2    EXCHANGEABLE PAIRS

In the context of the Ising model, the method of exchangeable pairs provides a powerful technique for obtaining nonasymptotic variance bounds for functions of the boolean hypercube. In the context

of the Ising model $\mu_{G,\beta}$, given a function $f : \{-1, 1\}^n \to \mathbb{R}$, to bound $\mathrm{Var}_\mu(f)$, the exchangeable pairs method constructs a pair $(\boldsymbol{\sigma}, \boldsymbol{\sigma}')$ such that $(\boldsymbol{\sigma}, \boldsymbol{\sigma}') \overset{d}{=} (\boldsymbol{\sigma}', \boldsymbol{\sigma})$ and the transition from $\boldsymbol{\sigma}$ to $\boldsymbol{\sigma}'$ is obtained via a single-site Glauber dynamics step (in our case truncated Glauber dynamics expanded on in the sequel).

Concretely, let $\boldsymbol{\sigma}'$ be obtained by resampling the spin at a uniformly chosen site $i \in V$ according to the conditional distribution $\mathbf{Pr}_{\Phi,\beta}(\cdot \mid \boldsymbol{\sigma}_{V\setminus\{i\}})$. Then $(\boldsymbol{\sigma}, \boldsymbol{\sigma}')$ is an exchangeable pair. Let $F(\boldsymbol{\sigma}, \boldsymbol{\sigma}') = \mathbb{E}[f(\boldsymbol{\sigma}) - f(\boldsymbol{\sigma}')|\boldsymbol{\sigma}_{-i}]$. The variance of $f$ can be bounded via

$$\mathrm{Var}_\mu(f) \leq \frac{1}{2}\mathbb{E}_\mu\left[(f(\boldsymbol{\sigma}) - f(\boldsymbol{\sigma}'))F(\boldsymbol{\sigma}, \boldsymbol{\sigma}')\right].$$

Under Lipschitz continuity with respect to the Hamming distance, this expression can be further bounded by quantities involving local influences and conditional variances, allowing for the control $\mathrm{Var}_\mu(f)$ in terms of the geometry of $G$ and the interaction strengths $A_{ij}$.

## B   ALGORITHM FOR MAXIMIZING THE PSEUDO-LIKELIHOOD

In this section, we present a polynomial-time algorithm for optimizing the pseudo-likelihood objective using projected gradient descent. To guarantee convergence to the optimum, we rely on the following lemma from (Bubeck et al., 2015).

**Lemma B.1** ((Bubeck et al., 2015) Theorem 3.10)**.** *Let $f$ be $\alpha-$strongly convex and $\lambda-$smooth on the convex set $\mathcal{X}$. Then projected gradient descent with step-size $\eta = 1/\lambda$, satisfies for $t \geq 0$,*

$$\|\mathbf{x}_{t+1} - \mathbf{x}^*\|_2^2 \leq \exp(\alpha t/\lambda)\|\mathbf{x}_1 - \mathbf{x}^*\|_2^2.$$

*Therefore, setting $R = \|\mathbf{x}_1 - \mathbf{x}^*\|_2$ and $t = 2(\lambda/\alpha)(\log(R) - \log(\epsilon))$ guarantees that $\|\mathbf{x}_t - \mathbf{x}^*\|_2 \leq \epsilon$.*

Given the $\Omega(n/\Delta^3)$-strong convexity of the pseudolikelihood function (Lemma 3.3) and the 1-Lipschitz continuity of its gradient, we apply projected gradient descent (PGD) with step size $\eta = 1$ to obtain a $1/\sqrt{n}$-accurate estimate of the MPLE. The algorithm is presented in Algorithm 1.

---

**Algorithm 1** Projected Gradient Descent

---

1: **Input:** Vector sample $\boldsymbol{\sigma}$, Magnetizations $m_i(\boldsymbol{\sigma}) = \sum_j A_{ij}\boldsymbol{\sigma}_j$, $k$-SAT Formula $\Phi = \Phi_{n,k,d}$

2: **Output:** Maximum Pseudolikelihood Estimate $\hat{\beta}$
3: Initialize: $\beta^0 = 0$, grad $= +\infty$, $\eta = 1$, flippable indices $\mathcal{F}(\boldsymbol{\sigma}) = \emptyset$
4: **for** $i$ in $\{1, ..., n\}$ **do**
5:     **if** $(-\sigma_i, \boldsymbol{\sigma}_{-i})$ is a satisfying assignment of $\Phi$ **then**
6:         $\mathcal{F}(\boldsymbol{\sigma}) \leftarrow \mathcal{F}(\boldsymbol{\sigma}) \cup \{i\}$
7:     **end if**
8: **end for**
9: $t \leftarrow 0$
10: **while** $|\text{grad}| > \frac{1}{\sqrt{n}}$ **do**
11:     grad $\leftarrow -\frac{1}{n}\sum_{i \in \mathcal{F}(\boldsymbol{\sigma})}\left[m_i(\boldsymbol{\sigma})(\sigma_i - \tanh(\beta^t m_i(\boldsymbol{\sigma})))\right]$
12:     $\beta^{t+1} \leftarrow \beta^t - \eta\text{grad}$
13:     $t \leftarrow t + 1$
14:     **if** $\beta^{t+1} < -B$ **then**
15:         $\beta^{t+1} \leftarrow -B$
16:     **end if**
17:     **if** $\beta^{t+1} > B$ **then**
18:         $\beta^{t+1} \leftarrow B$
19:     **end if**
20: **end while**
21: **return** $\beta_t$

---

## C  Omitted Proofs of Section 3.1 (Proof of Theorem 2)

In this section, we give a proof of Theorem 2, in the process demonstrating Lemma 3.1.

*Proof of Theorem 2.* Recall the first and second derivative bounds proved in Lemma 3.2 and Lemma 3.3, respectively

$$\mathbf{Pr}_{\beta^*}\left[|\phi_1(\beta^*;\boldsymbol{\sigma})| \leq \sqrt{\frac{(12+4B)n}{\delta}}\right] \geq 1-\delta, \quad \text{and } \mathbf{Pr}_{\beta^*}\left[\frac{\partial\phi^2(\beta;\boldsymbol{\sigma})}{\partial^2\beta} \geq \frac{n\exp(-B)}{\Delta^3(8kd)^2}\right] \geq 1-\frac{(24+8B)}{n^{0.1}}$$

The union bound implies that the event $\mathcal{A} = \{\boldsymbol{\sigma} \in \Omega(\Phi) : |\phi_1(\boldsymbol{\sigma};\beta^*)| \leq c\sqrt{n}, \min_{\beta\in(-B,B)} \phi_2(\boldsymbol{\sigma};\beta) \geq \Omega(n/\Delta^3)\}$ occurs with probability $1 - o(1)$.

To conclude the claim, we relate $\hat{\beta}$ with $\beta^*$, through smooth interpolation of both the parameter values themselves $\beta(t) = t\hat{\beta} + (1-t)\beta^*$, and the gradient $s(t) = (\hat{\beta} - \beta^*)\phi_1(\beta(t);\boldsymbol{\sigma})$. Via the chain rule, we notice that $s'(t) = (\hat{\beta} - \beta^*)^2\phi_2(\beta(t);\boldsymbol{\sigma})$. The fundamental theorem of calculus implies

$$-(\hat{\beta} - \beta^*)\phi_1(\beta;\boldsymbol{\sigma}) = s(1) - s(0) = \int_0^1 s'(t)dt = (\hat{\beta} - \beta^*)^2 \int_0^1 \phi_2(\beta(t);\boldsymbol{\sigma})dt$$

The log-pseudolikelihood is a convex objective, $\phi_2(\tilde{\beta};\boldsymbol{\sigma}) \geq 0$, for all $\tilde{\beta} \in (-B,B)$ and $\boldsymbol{\sigma} \in \mathcal{A}$ yielding,

$$|\hat{\beta} - \beta^*||\phi_1(\beta^*,\boldsymbol{\sigma})| \geq (\hat{\beta} - \beta^*)^2\left|\int_0^1 \phi_2(\beta(t);\boldsymbol{\sigma})dt\right| \geq (\hat{\beta} - \beta^*)^2 \min_{\tilde{\beta}\in(-B,B)} \phi_2(\tilde{\beta};\boldsymbol{\sigma}).$$

Rearranging this expression and using the fact that $\boldsymbol{\sigma} \in \mathcal{A}$,

$$|\hat{\beta} - \beta^*| \leq \frac{|\phi_1(\beta^*;\boldsymbol{\sigma})|}{\min_{\tilde{\beta}\in(-B,B)} \phi_2(\tilde{\beta};\boldsymbol{\sigma})} \leq \mathcal{O}\left(\frac{\Delta^3}{\sqrt{n}}\right), \text{ for all } \boldsymbol{\sigma} \in \mathcal{A}.$$

Recalling that $\boldsymbol{\sigma} \in \mathcal{A}$ with probability $1 - o(1)$ proves the desired claim. □

## D  Omitted Proofs in Section 3.2 (Proof of Lemma 3.2)

In this section, we provide a proof of the upper bound in probability for the first derivative of the log-pseudolikelihood objective, restated below for reference.

**Lemma 3.2.** *Fix a constant $\delta > 0$. The log-pseudolikelihood $\phi(\beta;\boldsymbol{\sigma})$ of a truncated Ising model fulfilling Assumption 1 satisfies the following upper bound in probability, for all $\beta \in \mathbb{R}$.*

$$\mathbf{Pr}_\beta\left[|\phi_1(\beta;\boldsymbol{\sigma})| \leq \sqrt{\frac{(12+4B)n}{\delta}}\right] \geq 1 - \delta.$$

*Proof.* To begin, we demonstrate our upper bound in probability over the first derivative of the log-pseudolikelihood $\phi(\beta;\boldsymbol{\sigma})$, showing this concentration inequality via the technique of exchangeable pairs introduced by (Chatterjee, 2007). Define the *anti-symmetric* function, $F : S \times S \to \mathbb{R}$,

$$F(\boldsymbol{\tau},\boldsymbol{\tau}') = \frac{1}{2}\sum_{i\in[n]}(m_i(\boldsymbol{\tau}) + m_i(\boldsymbol{\tau}'))(\tau_i - \tau_i')$$

Let $\boldsymbol{\sigma}$ drawn from the Ising model truncated by $\Phi$. We construct a new assignment $\boldsymbol{\sigma}'$, via taking one-step of the Glauber dynamics over the Markov random field induced by the Ising model; in other words, we select a coordinate $J \in [n]$ at random and fix $\boldsymbol{\sigma}'_{-J} := \boldsymbol{\sigma}_{-J}$ and redraw the remaining coordinate $\sigma'_J$ from the conditional distribution $\mathbf{Pr}_\beta(\cdot|\boldsymbol{\sigma}_{-J})$. The value of $F$ on $(\boldsymbol{\sigma},\boldsymbol{\sigma}')$ simplifies as,

$$F(\boldsymbol{\sigma},\boldsymbol{\sigma}') = m_J(\boldsymbol{\sigma})(\sigma_J - \sigma'_J).$$

Define the function $f(\boldsymbol{\sigma})$ as the *conditional* expectation of $F(\boldsymbol{\sigma}, \boldsymbol{\sigma}')$ with respect to $\boldsymbol{\sigma}$, that is

$$f(\boldsymbol{\sigma}) = \mathbb{E}_J \left( F(\boldsymbol{\sigma}, \boldsymbol{\sigma}') | \boldsymbol{\sigma} \right) = \frac{1}{n} \sum_{i \in [n]} m_i(\boldsymbol{\sigma})(\sigma_i - \mathbb{E}(\sigma_i | \boldsymbol{\sigma}_{-i}))$$

$$= \frac{1}{n} \sum_{i \in \mathcal{F}(\boldsymbol{\sigma})} m_i(\boldsymbol{\sigma})(\sigma_i - \tanh(\beta m_i(\boldsymbol{\sigma}))$$

$$= -\frac{1}{n} \frac{\partial}{\partial \beta} \phi(\beta; \boldsymbol{\sigma})$$

To show prove the desired result, it suffices to show a bound on the second moment of $f(\boldsymbol{\sigma})$. Observe that $(\boldsymbol{\sigma}, \boldsymbol{\sigma}')$ is indeed an exchangeable pair as

$$\mathbb{E}_\beta[f(\boldsymbol{\sigma})^2] = \mathbb{E}_{\beta,J}[f(\boldsymbol{\sigma})F(\boldsymbol{\sigma}, \boldsymbol{\sigma}')] = \mathbb{E}_{\beta,J}[f(\boldsymbol{\sigma}')F(\boldsymbol{\sigma}', \boldsymbol{\sigma})].$$

Moreover, the anti-symmetric nature of $F(\boldsymbol{\sigma}, \boldsymbol{\sigma}')$ implies $\mathbb{E}_{\beta,J}[f(\boldsymbol{\sigma}')F(\boldsymbol{\sigma}', \boldsymbol{\sigma})] = -\mathbb{E}_{\beta,J}[f(\boldsymbol{\sigma}')F(\boldsymbol{\sigma}, \boldsymbol{\sigma}')]$. These facts combine to recast $\mathbb{E}_\beta[f(\boldsymbol{\sigma})]$ as follows,

$$\mathbb{E}_\beta \left[ f(\boldsymbol{\sigma})^2 \right] = \mathbb{E}_{\beta,J}[f(\boldsymbol{\sigma})F(\boldsymbol{\sigma}, \boldsymbol{\sigma}')] = -\mathbb{E}_{\beta,J}[f(\boldsymbol{\sigma}')F(\boldsymbol{\sigma}, \boldsymbol{\sigma}')]$$

$$= \frac{1}{2} \mathbb{E}_{\beta,J} \left[ (f(\boldsymbol{\sigma}) - f(\boldsymbol{\sigma}'))F(\boldsymbol{\sigma}, \boldsymbol{\sigma}') \right]$$

If $\boldsymbol{\sigma} = \boldsymbol{\sigma}'$ then this expression is rendered trivially zero, and hence we need only analyse the case when $\sigma_I' = -\sigma_I$. If the redrawn coordinate $I$ is selected from the set of flippable indices, this probability is,

$$p_i(\boldsymbol{\sigma}) := \frac{\exp(-\sigma_i \beta m_i(\boldsymbol{\sigma}))}{\exp(-\beta m_i(\boldsymbol{\sigma})) + \exp(\beta m_i(\boldsymbol{\sigma}))} = \mathbf{Pr}(\sigma_i' = -\sigma_i | \boldsymbol{\sigma}, I = i, i \in \mathcal{F}(\boldsymbol{\sigma}))$$

and when $I \notin \mathcal{F}(\boldsymbol{\sigma})$ this probability is zero. Using the definitions of $f(\boldsymbol{\sigma})$ and $F(\boldsymbol{\tau}, \boldsymbol{\tau}')$ above, this expression is simplified as follows, where $\boldsymbol{\sigma}^{(i)} = (-\sigma_i, \boldsymbol{\sigma}_{-i})$.

$$\frac{1}{2} \mathbb{E}_J \left[ (f(\boldsymbol{\sigma}) - f(\boldsymbol{\sigma}'))F(\boldsymbol{\sigma}, \boldsymbol{\sigma}') | \boldsymbol{\sigma} \right] = \frac{1}{n} \sum_{i \in \mathcal{F}(\boldsymbol{\sigma})} (f(\boldsymbol{\sigma}) - f(\boldsymbol{\sigma}^{(i)}))F(\boldsymbol{\sigma}, \boldsymbol{\sigma}^{(i)})p_i(\boldsymbol{\sigma})$$

$$= \frac{1}{n} \sum_{i \in \mathcal{F}(\boldsymbol{\sigma})} (f(\boldsymbol{\sigma}) - f(\boldsymbol{\sigma}^{(i)}))m_i(\boldsymbol{\sigma})(\sigma_i - \tanh(\beta m_i(\boldsymbol{\sigma}))p_i(\boldsymbol{\sigma})$$

$$:= \frac{1}{n} \sum_{i \in \mathcal{F}(\boldsymbol{\sigma})} T_{1i}T_{2i}$$

*Bound on $T_{1i}$:* We now bound each of term in the above expression, beginning with $T_{1i}$ where $i$ is flippable. The Taylor expansion of $f(\boldsymbol{\sigma}^{(i)})$ centered at $f(\boldsymbol{\sigma})$ yields,

$$|f(\boldsymbol{\sigma}^{(i)}) - f(\boldsymbol{\sigma})| \leq |\sigma_i - \sigma_i^{(i)}| \max_{w \in [-1,1]} \frac{\partial f}{\partial \sigma_i}((w, \boldsymbol{\sigma}_{-i})) = \max_{w \in [-1,1]} 2 \cdot \frac{\partial f}{\partial \sigma_i}((w, \boldsymbol{\sigma}_{-i})),$$

where $w$ is point along the line with endpoints $\boldsymbol{\sigma}$ and $\boldsymbol{\sigma}^{(j)}$.

The partial derivative of $f(\boldsymbol{\sigma})$ with respect to $\sigma_i$ evaluated at a spin configuration $\boldsymbol{\tau} \in S$ is

$$\frac{\partial f}{\partial \sigma_i}(\boldsymbol{\tau}) = \frac{1}{n} \sum_{j \in \mathcal{F}(\boldsymbol{\tau})} \left( \left( \mathbf{1}_{i=j} - \frac{\beta A_{ji}}{\cosh^2(\beta m_i(\boldsymbol{\tau}))} \right) m_j(\boldsymbol{\tau}) + (\tau_j - \tanh(\beta m_j(\boldsymbol{\tau}))) \frac{\partial m_j(\boldsymbol{\tau})}{\partial \sigma_i} \right)$$

The assumptions on $G$ implies $|m_i(\boldsymbol{\tau})| \leq 1$ for all values of $i \in I$. Furthermore, $|\cosh(\cdot)| \geq 1$, yielding the following bound on the rescaled first term.

$$\left| \sum_{j \in \mathcal{F}(\boldsymbol{\tau})} \left( \mathbf{1}_{i=j} - \frac{\beta A_{ji}}{\cosh^2(\beta m_i(\boldsymbol{\tau}))} \right) m_j(\boldsymbol{\tau}) \right| \leq \left( |m_i(\boldsymbol{\tau})| + \sum_{\{j \neq i | j \in \mathcal{F}(\boldsymbol{\tau})\}} |\beta A_{ji}m_j(\boldsymbol{\tau})| \right)$$

Likewise, $\frac{\partial m_j(\boldsymbol{\sigma})}{\partial \sigma_i} = A_{ji}$ implies a bound on the second term.

$$\left| (\tau_j - \tanh(\beta m_j(\boldsymbol{\tau}))) \frac{\partial m_j(\boldsymbol{\tau})}{\partial \sigma_i} \right| \leq |(\tau_j - \tanh(\beta m_j(\boldsymbol{\tau})))| \left| \frac{\partial m_j(\boldsymbol{\tau})}{\partial \sigma_i} \right| \leq 2|A_{ji}|$$

Combining these two bounds yields

$$
\begin{aligned}
|T_{i1}| &\leq \max_{w \in [-1,1]} \left| \frac{\partial f}{\partial \sigma_i}((w, \boldsymbol{\sigma}_{-i})) \right| \\
&\leq \max_{w \in [-1,1]} \frac{1}{n} \left( |m_i((w, \boldsymbol{\sigma}_{-i}))| + \sum_{\{j \neq i | j \in \mathcal{F}(\boldsymbol{\sigma})\}} |\beta A_{ji} m_j((w, \boldsymbol{\sigma}_{-i}))| + 2|A_{ji}| \right) \\
&\leq \frac{1}{n} \left( 1 + \sum_{\{j \neq i | j \in \mathcal{F}(\boldsymbol{\sigma})\}} |A_{ji}|(2 + B) \right) \\
&\leq \frac{(6 + 2B)}{n}
\end{aligned}
$$

*Bound on $T_{2i}$:* Recall $|m_i(\boldsymbol{\sigma})| \leq 1$ for all $i \in I$ and $\boldsymbol{\sigma} \in \{-1,1\}^{|V/I|}$ and $|\tanh(x)| \leq 1, \forall x \in \mathbb{R}$. Then

$$
|T_{2i}| = |m_i(\boldsymbol{\sigma})(\sigma_i - \tanh(\beta m_i(\boldsymbol{\sigma}))p_i(\boldsymbol{\sigma})| \leq 2
$$

*Putting together the pieces:* We are now ready to construct our final bound on $\mathbb{E}_\beta(f(\boldsymbol{\sigma})^2)$.

$$
\begin{aligned}
\mathbb{E}_\beta(f(\boldsymbol{\sigma})^2) &= \frac{1}{2} \mathbb{E}_{\beta,J} \left( (f(\boldsymbol{\sigma}) - f(\boldsymbol{\sigma}'))F(\boldsymbol{\sigma}, \boldsymbol{\sigma}') \right) \\
&= \frac{1}{2n} \mathbb{E}_\beta \left( \sum_{i \in I} T_{1i} T_{2i} e_i(\boldsymbol{\sigma}) \right) \\
&\leq \frac{1}{2n} \left( \sum_{i=1}^n \frac{(12 + 4B)}{n} \right) \\
&= \frac{(6 + 2B)}{n}
\end{aligned}
$$

Recalling the relationship between $f(\boldsymbol{\sigma})$ and $\frac{\partial \phi}{\partial \beta}$, the claim follows directly. $\qquad \square$

# E  OMITTED PROOFS OF SECTION 3.3 (STRONG CONVEXITY OF THE PSEUDO-LIKELIHOOD)

The primary aim of this section is proving Lemma 3.3, recounted here for convenience.

**Lemma 3.3.** *The log-pseudolikelihood $\phi(\beta; \boldsymbol{\sigma})$ of a Ising model, truncated by a $k$-SAT formula with $k \geq \frac{4\Delta^3(1 + \log(d^2 k + 1))}{\log(1 + \exp(-2B))}$, fulfilling Assumption 1 satisfies the following lower bound in probability for all $\beta \in (-B, B)$,*

$$
\mathbf{Pr}_{\beta^*} \left[ \frac{\partial \phi^2(\beta; \boldsymbol{\sigma})}{\partial^2 \beta} \geq \frac{n \exp(-B)}{\Delta^3 (8kd)^2} \right] \geq 1 - \frac{(24 + 8B)}{n^{0.1}}.
$$

Towards this goal, we provide a proof of Lemma 3.4 in Section E.1, Lemma 3.5 in Section E.2, and Lemmas 3.6 & 3.7 in Section E.3 before concluding Lemma 3.3 in Section E.4.

## E.1  PROOF OF LEMMA 3.4

The proof of this lemma proceeds by defining an explicit, algorithmic correspondence between bijections $\rho : V \to [n]$ and independent sets $I$. This mapping induces a measure $\mu_\rho$ over bijections, which in turn defines a distribution over the resulting independent sets. We analyze this distribution to bound the probability that a randomly generated independent set contains fewer than $\lambda k$ elements in some clause—a "bad event." Applying the Lovász Local Lemma, we show that with positive probability, none of these bad events occur, implying the existence of an independent set that satisfies the desired clause-wise coverage property.

### E.1.1 THE ALGORITHM

Given a bijection $\rho : V \to [n]$, we provide a simple algorithm for finding an independent set detailed formally below. To bound the probability of bad events, that fewer than $\lambda k$ elements from a clause are included in the independent set under the uniform measure over bijections $\mu_\rho$, we must ensure that the inclusion of distant vertices into the independent set occurs in a *independent* manner. This form of spatial independence is crucial for applying the Lovász Local Lemma, and it is established in the following lemma.

---

**Algorithm 2** Independent Set in Graph Based on Random Ordering

---

**Require:** Graph $G = (V, E)$ with $V = [n]$
**Ensure:** Set $S$ of selected vertices
 1: Sample a random permutation $\rho : V \to [n]$
 2: **function** INDEDGESET($\rho$)
 3:     Initialize $S \leftarrow \emptyset$
 4:     **for all** $e \in V$ **do**
 5:         **if** $\rho(u) > \max_{v \in N(u)} \rho(v)$ **then**
 6:             $S \leftarrow S \cup \{u\}$
 7:         **end if**
 8:     **end for**
 9:     **return** $S$
10: **end function**
11: **return** IndEdgeSet($\rho$)

---

**Lemma E.1.** *Fix two vertices $u, v \in V$ such $d(u, v) \geq 3$, over the graph $G$ induced by $A$. Over the uniform measure of orderings $\rho : V \to [n]$, $\mu_\rho$, the indicator random variables $\mathbf{1}\{u \text{ belongs to } \mathsf{IndEdgeSet}(\rho)\}$ and $\mathbf{1}\{v \text{ belongs to } \mathsf{IndEdgeSet}(\rho)\}$ are independent, that is*

$$\mathbf{Pr}_{\mu_\rho}[\{u, v \in \mathsf{IndEdgeSet}(\rho)\}] = \mathbf{Pr}_{\mu_\rho}[\{u \in \mathsf{IndEdgeSet}(\rho)\}] \cdot \mathbf{Pr}_{\mu_\rho}[\{v \in \mathsf{IndEdgeSet}(\rho)\}]$$

*Moreover, the probability a given node $v$ lies in $I$ is*

$$\mathbf{Pr}_{\mu_\rho}[\{v \in \mathsf{IndEdgeSet}(\rho)\}] \leq \frac{1}{\Delta + 1}$$

*Proof.* The event that a vertex $w \in V$ lies in $\mathsf{IndEdgeSet}(\rho)$ depends fundamentally on the structure of the bijection $\rho$. Specifically,

$$\{w \in \mathsf{IndEdgeSet}(\rho)\} = \left\{ \rho(w) > \max_{i \in N(w)} \rho(i) \right\},$$

where $N(w)$ denotes the neighbors of $w$ in the graph.

The joint probability, under the uniform measure $\mu_\rho$ over all bijections $\rho$, that two distinct vertices $u$ and $v$ both belong to the set $I = \mathsf{IndEdgeSet}(\rho)$,

$$\mathbf{Pr}_{\mu_\rho}[u \in I \text{ and } v \in I] = \mathbf{Pr}_{\mu_\rho}\left[ \rho(u) > \max_{w \in N(u)} \rho(w), \ \rho(v) > \max_{w' \in N(v)} \rho(w') \right],$$

depends only on the relative ordering of the values of $\rho$ on the set $\{u, v\} \cup N(u) \cup N(v)$, which has size at most $2\Delta + 2$. Moreover, since $d(u, v) \geq 3$, the neighborhoods $N(u)$ and $N(v)$ are disjoint.

Towards computing this probability, observe that, there are $|N(u)|!$ permutations of $\{u\} \cup N(u)$ in which $u$ appears first, and similarly there are $|N(v)|!$ permutations of $\{v\} \cup N(v)$ in which $v$ appears first. All orderings over $\{u \cup v \cup N(u) \cup N(v)\}$, which place $u, v$ first among their respective neighbors are shuffles of existing orderings of $\{u \cup N(u)\}$ and $\{v \cup N(v)\}$. Counting combinations, there are $\binom{|N(u)| + |N(v)| + 2}{|N(u)| + 1}$ ways to interleave the two sets $\{u\} \cup N(u)$ and $\{v\} \cup N(v)$ while preserving their internal orderings, implying the number of permuations satisfying the above condition is:

$$\binom{|N(u)| + |N(v)| + 2}{|N(u)| + 1} \cdot |N(u)|! \cdot |N(v)|! = \frac{(|N(u)| + |N(v)| + 2)!}{(|N(u)| + 1)(|N(v)| + 1)}.$$

As the total number of permutations of the relevant elements is $(|N(u)| + |N(v)| + 2)!$, the joint probability is:

$$\mathbf{Pr}_{\mu_\rho}[u, v \in \mathsf{IndEdgeSet}(\rho)] = \frac{1}{(|N(u)| + 1)(|N(v)| + 1)}.$$

A similar argument yields that for a single vertex $v$ there are $N(v)!/(N(v) + 1)! = 1/(N(v) + 1)$ permutations placing it first in relative order among its neighbors. This implies the probability over the uniform measure over bijections that $v$ belongs to the induced independent set is:

$$\mathbf{Pr}_{\mu_\rho}[v \in \mathsf{IndEdgeSet}] = \mathbf{Pr}_{\mu_\rho}\left[\rho(v) > \max_{j \in N(v)} \rho(j)\right] = \frac{1}{|N(v)| + 1}.$$

The desired conclusion follows by combining the expressions for the single and joint probabilities.

$\square$

In advance of proving Lemma 3.4, we introduce an important tool that relates an arbitrary collection of potentially correlated random variables to independently and identically distributed variables, which will enable the use of Chernoff bounds in the sequel.

**Lemma E.2** ((Frieze & Karoński, 2015) *Section 23.9*). *Suppose that $\{Y_i\}_{i \in [n]}$ are independent random variables and that $\{X_i\}_{i \in [n]}$ are random variables so that for any real $t$ and $i \in [n]$, it holds that*

$$\mathbf{Pr}[X_i \geq t | X_1, \ldots X_{i-1}] \geq \mathbf{Pr}[Y_i \geq t].$$

*Then, for any real $t$,*

$$\mathbf{Pr}[X_1 + \ldots + X_n \geq t] \geq \mathbf{Pr}[Y_1 + \ldots + Y_n \geq t].$$

Armed with this background, we now prove Lemma 3.4, recounted here for reference.

**Lemma 3.4.** *Let $G$ be a graph with maximum degree $\Delta = o(n^{1/6})$ and $\Phi$ be a $k$-SAT formula. If $k > 10\Delta^3(1 + \log(dk\Delta^2))$, then there exists an independent set $I \subset V$ such that $\Phi'$, $\Phi$ truncated on $V \setminus I$, is a $\lambda k$- SAT formula where $\lambda = 1/4\Delta^3$.*

*Proof.* To begin, consider a clause $C \in \mathcal{C}$, and select a maximal collection of *neighborhood* disjoint variables $C' \subseteq C$. In other words, we require that for all $i, j \in C'$, $d(i, j) \geq 3$. The maximum size of a two-hop neighborhood of a given point is at most $\Delta^2 + 1$, implying the size of $C'$ is at least $k/(\Delta^2 + 1)$. Denote the function $f_C(\rho) = |\{\mathsf{IndEdgeSet}(\rho) \cap C\}|$. As each pair of elements in $C'$ is at least distance 3 apart from each other, Lemma E.1 implies the following bound on the expectation of $f_C(\rho)$.

$$\mathbb{E}_{\mu_\rho}[f_C(\rho)] \geq \mathbb{E}_{\mu_\rho}[f_{C'}(\rho)] = \sum_{v_i \in C'} \mathbf{Pr}_{\mu_\rho}[v_i \in I] \geq \frac{k}{(\Delta + 1)(\Delta^2 + 1)}$$

Moreover, this directly implies that for all $t$, that for $Y_i \sim \mathrm{Bern}(1/(\Delta + 1))$

$$\mathbf{Pr}[\mathbf{1}\{v_i \in I\} | \mathbf{1}\{v_1 \in I\}, \ldots, \mathbf{1}\{v_{i-1} \in I\}] \geq \mathbf{Pr}[Y_i \geq t].$$

Given this information, we use Chernoff bounds to find an upper bound on the event $\mathbf{Pr}[Y_1 + \cdots + Y_n \geq k/(2(\Delta^2 + 1)(\Delta + 1))]$, and use this to in turn bound $\mathbb{E}_{\mu_\rho}[f_C(\rho)]$.

$$\mathbf{Pr}_{\mu_\rho}\left[f_C(\rho) < \frac{k}{(2(\Delta + 1)(\Delta^2 + 1))}\right] < \mathbf{Pr}\left[\sum_{v_i \in C} Y_{v_i} < \frac{k}{(2(\Delta + 1)(\Delta^2 + 1))}\right]$$

$$\leq \mathbf{Pr}\left[\sum_{v_i \in C'} Y_{v_i} < \frac{k}{(2(\Delta + 1)(\Delta^2 + 1))}\right]$$

$$= \mathbf{Pr}\left[\sum_{v_i \in C'} Y_{v_i} - \mathbb{E}\left[\sum_{v_i \in C'} Y_{v_i}\right] \leq (1 - 1/2)\mathbb{E}\left[\sum_{v_i \in C'} Y_{v_i}\right]\right]$$

$$\leq \exp\left(-\frac{k}{8(\Delta + 1)(\Delta^2 + 1)}\right)$$

To construct a final bound on $k$ in terms of $d$ and $\Delta$ to ensure that a marking satisfying our desired conditions exists, we use the symmetric version of the Lovascz Local Lemma. Each variable appears in at most $d$ other clauses, and the event of selection into the independent set relies on its $\Delta^2 + 1$ neighbors which lie in its two-hop neighborhood. This implies the degree of the dependency graph of $\Phi$ is $kd(\Delta^2 + 1)$.

$$2e \exp\left(-\frac{k}{8(\Delta^2 + 1)(\Delta + 1)}\right)(kd(\Delta^2 + 1)) < 1$$

$$\exp\left(-\frac{k}{8(\Delta^2 + 1)(\Delta + 1)}\right) < \frac{1}{2e(kd(\Delta^2 + 1))}$$

$$-\frac{k}{8(\Delta^2 + 1)(\Delta + 1)} < -1 - \log(kd(\Delta^2 + 1)))$$

$$k > 8(1 + \log(kd) + 3\log(\Delta))(\Delta^2 + 1)(\Delta + 1)$$

The independent set $I$ induces a smaller CNF, $\Phi'_{n,k',d'} = (\mathcal{V}', \mathcal{C}')$, with clauses $\mathcal{C}' = \bigcup_{C \in \mathcal{C}} C \cap I$, each of which has at least $k/(2\Delta^3 + 2\Delta)$ variables, concluding the desired claim. □

### E.2 PROOF OF LEMMA 3.5

In this section, our aim to show Lemma 3.5. We accomplish this goal in two steps, **first** demonstrating each variable outside the independent set is flippable with probability $1 - \delta$, where $\delta \in (0, 1)$ and **second** demonstrating each variable *inside* the independent set is flippable with probability at least $1/2$.

Proceeding, we introduce the version of the Lovascz Local Lemma that will be used in bound of $s_i(\boldsymbol{\sigma})$.

**Lemma E.3** ((Guo et al., 2019)). *Suppose that $\mu(\boldsymbol{\sigma})$ is a product distribution over $\boldsymbol{\sigma}' \in \{-1, 1\}^k$. Let $A_i$ be an event determined by the elements of $\boldsymbol{\sigma}$, and denote $B(S) = \wedge_{i \in S} \bar{A}_i$. Then if there exists a vector $x$ such that $x \in (0, 1]^m$ and*

$$\mathbf{Pr}(A_i) \leq x_i \prod_{(i,j) \in E} (1 - x_j)$$

*then*

$$\mathbf{Pr}(B(S)) \geq \prod_{i \in S} (1 - x_i) > 0$$

*Moreover, let $E$ be an event determined by some of the coordinates of $\boldsymbol{\sigma}$ and let $\Gamma(E) = \{i \in S : var(A_i) \cap var(E) \neq \emptyset\}$. We then see that*

$$\mathbf{Pr}_\mu(E|B(S)) \leq \mathbf{Pr}_\mu(E) \prod_{i \in \Gamma(E) \cap S} (1 - x_i)^{-1}$$

**Lemma E.4.** *Given a sample $\boldsymbol{\sigma} \sim \mathbf{Pr}_{\beta,S}$, where $\beta \in (-B, B)$ and that the $k$-SAT formula $\Phi$ which induces the truncation set $S$, satisfies the following clause size bound*

$$k \geq \frac{4\Delta^3(1 + \log(d^2 k + 1))}{\log(1 + \exp(-2B))}.$$

[3] *Then for $\Delta \geq 5$, there exists an independent set $I$ following Lemma 3.4 such that*

$$\mathbf{Pr}_{\beta,S}[s_j(\boldsymbol{\sigma}) = 1] \geq 1/2 \quad \forall j \in V \setminus I.$$

*Proof.* Towards establishing the desired claim, we first need a bound on $k'$ with respect to $d$ to ensure the use of Lemma E.3 (the asymmetric LLL). For any assignment $\boldsymbol{\tau} \in \{-1, 1\}^{|V/I|}$, define $\Phi^\tau = (\mathcal{V}^\tau, \mathcal{C}^\tau)$ as the CNF formula obtained via truncation on the partial assignment $\boldsymbol{\tau}$, that is the assignment that removes clauses satisfied by $\boldsymbol{\tau}$ and removes literals from $\boldsymbol{\tau}$ from the remaining

---

[3]This term scales as $\gtrsim e^{2B}\Delta^3 \log(d^2 k)$.

clauses. Notice, the set of clauses $\mathcal{C}'$ within $\Phi'$ are merely the union of all clauses $\mathcal{C}^\tau$ over all $\Phi^\tau$, that is

$$\mathcal{C}' = \bigcup_{\tau \in \{-1,1\}^{|V/I|}} \mathcal{C}^\tau,$$

implying a bound that would guarantee a satisfying assignment for $\Phi'$, would in turn ensure the existence of the satisfiability of all $\Phi^\tau$.

Moreover, recall for any independent set $I$, conditioned on the variables outside of the independent set $V/I$, the Ising model collapses into a product distribution.

$$\mathbf{Pr}_\beta[\boldsymbol{\sigma}_I | \boldsymbol{\sigma}_{V/I}] = \prod_{i \in I} \frac{\exp\left(\beta \sum_{j \in V/I} A_{ij} \sigma_i \sigma_j\right) e_i(\boldsymbol{\sigma})}{\exp\left(\beta \sum_{j \in V/I} A_{ij} \sigma_j\right) + \exp\left(-\beta \sum_{j \in V/I} A_{ij} \sigma_j\right)}$$

This directly implies

$$\min_{\kappa \in \{-1,1\}} \mathbf{Pr}_\beta[\sigma_i = \kappa | e_i(\boldsymbol{\sigma}), \boldsymbol{\sigma}_{V/I}] \geq \frac{\exp(|\beta|)}{\exp(\beta\Delta) + \exp(-\beta\Delta)} = \frac{\exp(2|\beta|)}{1 + \exp(2|\beta|)}$$

For each clause $C' \in \mathcal{C}'$, the event $\{C'$ is not satisfied$\}$ depends on $dk$ variables which lie in at most $d^2k$ clauses. Following the setup of Lemma E.3, we set $x(C') = 1/(D+1)$, $D := d^2k$, and notice if $k' \geq \frac{1+\log(d^2k+1)}{\log(1+\exp(-2|\beta|))}$

$$\left(\frac{\exp(2|\beta|)}{\exp(2|\beta|) + 1}\right)^{k'} \leq \left(\frac{D}{D+1}\right)^D \frac{1}{D+1}$$

$$\left(\frac{D}{D+1}\right)^{-D} (D+1) \leq (1 + \exp(-2|\beta|))^{k'}$$

$$e(d^2k + 1) \leq (1 + \exp(-2|\beta|))^{k'}$$

$$1 + \log(d^2k + 1) \leq k' \log(1 + \exp(-2|\beta|))$$

$$\frac{1 + \log(d^2k + 1)}{\log(1 + \exp(-2|\beta|))} \leq k'$$

Counting combinations, there is only one way to assign all the variable in $C'$ such that the clause is not satisfying. The worst case probability a clause $C'$ takes any given configuration under the *untruncated* distribution $\mu_\beta$ is at most

$$\mathbf{Pr}_{\mu_\beta}[\text{clause } C' \text{ is not satisfied}] \leq \left(\frac{\exp(2|\beta|)}{1 + \exp(2|\beta|)}\right)^{k'}.$$

For every pinning $\tau \in \{-1, 1\}^{|V/I|}$, we can equivalently find a upper bound for the probability that $\{s_i(\boldsymbol{\sigma}) = 0\}$, under the conditional distribution, i.e. given $\boldsymbol{\sigma} \in S_\tau$, where $S_\tau$ is the set of satisfying assignments of $\Phi$ where $\boldsymbol{\sigma}_{V/I}$ is pinned to $\tau$.

$$\mathbf{Pr}_\beta[s_i(\boldsymbol{\sigma}) = 0] = \sum_{\tau \in \{-1,1\}^{|V/I|}} \mathbf{Pr}_{\mu_\beta}[s_i(\boldsymbol{\sigma}) = 0 | \boldsymbol{\sigma} \in S_\tau] \mathbf{Pr}_\beta[\boldsymbol{\sigma} \in S_\tau]$$

$$\leq \max_{\tau \in \{-1,1\}^{|V/I|}} \mathbf{Pr}_{\mu_\beta}[s_i(\boldsymbol{\sigma}) = 0 | \boldsymbol{\sigma} \in S_\tau]$$

$$\leq \mathbf{Pr}_{\mu_\beta}[s_i(\boldsymbol{\sigma}) = 0] \left(1 - \frac{1}{D+1}\right)^{-D} \leq ed \left(\frac{\exp(2|\beta|)}{1 + \exp(2|\beta|)}\right)^{k'}$$

The final inequality derives from our use of Lemma E.3, relating probabilities between the truncated and untruncated distributions. We lastly rearrange for $k'$ to derive the final result.

$$\mathbf{Pr}_\beta[s_i(\boldsymbol{\sigma})] = (1-\delta) \geq 1 - ed\left(\frac{\exp(2|\beta|)}{1+\exp(2|\beta|)}\right)^{k'}$$

$$ed\left(\frac{\exp(2|\beta|)}{1+\exp(2|\beta|)}\right)^{k'} \leq \delta$$

$$1 + \log(d) - k'(\log(1+\exp(2|\beta|)) - 2|\beta|) \geq \log(\delta)$$

$$\frac{1 + \log(d) - \log(\delta)}{\log(1+\exp(2|\beta|)) - 2|\beta|} \leq k'$$

$\square$

Lastly, to guarantee there are a sufficient number of flippable variables in the independent set itself, we use the following result from (Galanis et al., 2024) to find a lower bound on the number of variables within the independent set that are flippable under a product distribution.

**Lemma E.5** (Lemma 15 & 16 (Galanis et al., 2024)). *Consider a formula $\Phi' = \Phi_{n,k',d}$ with $k' \geq \frac{2\log(dk')+\Theta(1)}{\lambda \log(1+e^{-\beta})}$, and an associated product measure $\mu_\gamma$ over the hypercube $\{-1,1\}^n$, such that each variable is set to 1 independently with probability $(\exp(\gamma))/(1+\exp(\gamma))$. Then for each variable $\sigma_i \in \mathcal{V}$,*

$$\mathbf{Pr}_\gamma[\sigma_i \text{ is not flippable}] \leq 1/2.$$

*Moreover, we can find a collection of $R \subseteq [n]$ with $|R| \geq n/(2kd)^2$ that are neighborhood disjoint in the interaction graph of the $k$-SAT formula $\Phi'$ such that for all subcollections $\{i_1, ..., i_t\} \subset R$,*

$$\mathbf{Pr}_\gamma\left[e_{i_t}(\boldsymbol{\sigma}) = 1 | e_{i_1}(\boldsymbol{\sigma}) = 1, ..., e_{i_{t-1}}(\boldsymbol{\sigma}) = 1\right] \geq 1/2.$$

*This in turn implies, with probability $1 - \exp(-\Omega(n))$ over the choice of $\boldsymbol{\sigma} \sim \mathrm{Pr}_{\phi,\beta}$, it holds that $\sum_{i \in R} e_i(\boldsymbol{\sigma}) \geq |R|/3$.*

The second half of Lemma 3.5 then follows from this corresponding results in (Galanis et al., 2024). To relate these results to our setting, observe that under our product distribution, the probability that any variable is set to one is at most $e^{-B}/(1+e^{-B})$, and the true marginals may in fact be *more* balanced. Consequently, the conditions of their result are satisfied in our regime.

### E.3    PROOF OF LEMMAS 3.6 & 3.7

For reference, recall the expression for the Hessian of the log pseudo-likelihood,

$$\phi_2(\beta; \boldsymbol{\sigma}) = \frac{\partial^2 \phi(\beta; \boldsymbol{\sigma})}{\partial \beta^2} = \sum_{i=1}^n \frac{m_i^2(\boldsymbol{\sigma})}{\cosh^2(\beta m_i(\boldsymbol{\sigma}))} e_i(\boldsymbol{\sigma}).$$

Towards demonstrating $\phi_2(\beta; \boldsymbol{\sigma}) \in \Omega(n)$ with probability $1 - o(1)$ for all $\beta \in (-B, B)$, we provide a lower bound of the conditional mean of the magnetization of the flippable variables via proving Lemma 3.6.

*Proof of Lemma 3.6.* For all $\boldsymbol{\sigma} \in \{-1,1\}^n$, consider $(\sum_{t \neq j} A_{it}\sigma_t + A_{ij})^2$ and $(\sum_{t \neq j} A_{it}\sigma_t - A_{ij})^2$. If $\sum_{t \neq j} A_{it}\sigma_t$ and $A_{ij}$ have the same sign, the first term is at least $A_{ij}^2$ and the opposite sign,

the second is at least $A_{ij}^2$. This implies that

$$
\begin{aligned}
\mathbb{E}_{\beta^*}[m_i^2(\boldsymbol{\sigma})|\boldsymbol{\sigma}_{-j}] &= \sum_{\kappa \in \{0,1\}} \mathbb{E}_{\beta^*}[m_i^2(\boldsymbol{\sigma})|e_j(\boldsymbol{\sigma}) = \kappa]\mathbf{Pr}_\beta[e_j(\boldsymbol{\sigma}) = \kappa] \\
&\geq \sum_{\kappa \in \{0,1\}} A_{ij}^2 \cdot \min_{\ell \in \{-1,1\}}(\mathbf{Pr}_{\beta^*}[\sigma_j = \ell|\boldsymbol{\sigma}_{-j}, e_j(\boldsymbol{\sigma}) = \kappa])\mathbf{Pr}_\beta[e_j(\boldsymbol{\sigma}) = \kappa] \\
&\geq \frac{A_{ij}^2}{2}\exp\left(-\left|\beta^*\sum_{t \neq j} A_{jt}\sigma_j\right|\right)\mathbf{Pr}_{\beta^*}[e_j(\boldsymbol{\sigma}) = 1] \\
&\geq \frac{A_{ij}^2}{2}\exp(-B)\mathbf{Pr}_{\beta^*}[e_j(\boldsymbol{\sigma}) = 1]
\end{aligned}
$$

$\square$

This result provides a lower bound for $\phi_2(\beta; \boldsymbol{\sigma})$ in terms of scaled elements of the interaction matrix $A_{ij}$. To maximize this lower bound, we wish to select columns $h(i)$ for each row to ensure the value of $A_{ih(i)}$ is as large as possible. To this end, consider an injective mapping $h : V \to V$. The requirement that $\|A\|_\infty \leq 1$ and $\|a_i\|_2 \geq c$, implies the existence of a edge $A_{ij}$ for each row such that $A_{ij} > c'$. Moreover, due to the connectivity of the graph, we can select a subset $I' \subset I$ of size at least $|I|/\Delta > n/\Delta^2$, with an *unique* neighbor $h(i) \in V/I$, where $A_{ih(i)} \geq c'$. Outside of $I'$, we assign partners arbitrarily making sure to keep $h(i)$ a bijection. Towards this goal, we present a proof of Lemma 3.7.

*Proof of Lemma 3.7.* We begin by constructing a set $R \subset I$ of variables that are disjoint in both the incidence graph of the $k-$SAT formula and the graph $G$. A simple greedy algorithm that selects a point arbitrarily, deletes its 2-hop neighbors in both graphs and recurses has size at least $n/(2k'd)^2\Delta^2$. This implies the sum of conditional magnetizations takes the following form.

$$
\begin{aligned}
\sum_{i \in R} \mathbb{E}_{\beta^*}[m_i^2(\boldsymbol{\sigma})e_i(\boldsymbol{\sigma})|\boldsymbol{\sigma}_{-h(i)}] &= \sum_{i \in R} \mathbb{E}_{\beta^*}[m_i^2(\boldsymbol{\sigma})|\boldsymbol{\sigma}_{-h(i)}]\mathbf{Pr}_{\beta^*}[e_i(\boldsymbol{\sigma}) = 1|e_{i_1}(\boldsymbol{\sigma}) = 1, ..., e_{i_{t-1}}(\boldsymbol{\sigma}) = 1] \\
&\geq \frac{1}{2}\sum_{i \in R} \mathbb{E}_{\beta^*}[m_i^2(\boldsymbol{\sigma})|\boldsymbol{\sigma}_{-h(i)}] \\
&\geq \sum_{t \in |R|} \frac{\exp(-B)}{4}\mathbf{Pr}_{\beta^*}[e_{h(i)}(\boldsymbol{\sigma}) = 1] \\
&\geq \frac{n\exp(-B)(1-\delta)}{\Delta(4kd\Delta)^2}
\end{aligned}
$$

Note the last inequality comes from Lemma 15, and the fact that $s_i(\boldsymbol{\sigma}) \leq e_i(\boldsymbol{\sigma})$. $\square$

### E.4 PROOF OF LEMMA 3.3

Armed with the tools from the previous section we now prove Lemma 3.3.

*Proof of Lemma 3.3.* We begin by expanding out $m_i(\boldsymbol{\sigma})$ into its component parts, namely

$$
m_i^2(\boldsymbol{\sigma}) = \left(\sum_{j \neq h(i)} A_{ij}\sigma_j\right)^2 + A_{ih(i)}^2 + 2\left(\sum_{j \neq h(i)} A_{ij}\sigma_j\right)A_{ih(i)}\sigma_{h(i)}
$$

Cancelling common factors implies that

$$
\mathbb{E}_{\beta^*}\left[\left(2\sum_{(i,j)\in\mathcal{F}(\boldsymbol{\sigma})}\left(\sum_{j \neq h(i)} A_{ij}\sigma_j\right)A_{ih(i)}\left(\sigma_{h(i)} - \mathbb{E}_{\beta^*}[\sigma_{h(i)}|\boldsymbol{\sigma}_{h(i)}]\right)\right)^2\right].
$$

We merely sum over the flippable indices, as when $\sigma_{h(i)}$ is not flippable, the term $\sigma_{h(i)} - \mathbb{E}_{\beta^*}[\sigma_{h(i)}|\boldsymbol{\sigma}_{-h(i)}]$ collapses. Denoting $y_{it}(\boldsymbol{\sigma}) = 2\left(\sum_{j\neq t} A_{ij}\sigma_j\right)^2 A_{it}$ and recalling $\mathbb{E}_{\beta^*}[\sigma_i|\boldsymbol{\sigma}_{-i}] = \tanh(\beta^* m_i^2(\boldsymbol{\sigma}))$, yields the following simplified version of the above expression.

$$\mathbb{E}_{\beta^*}\left[\sum_{(i,j)\in\mathcal{F}(\boldsymbol{\sigma})}\left(y_{ih(i)}(\boldsymbol{\sigma})\left(\sigma_{h(i)} - \tanh(\beta^* m_{h(i)}(\boldsymbol{\sigma}))\right)^2\right)\right].$$

We aim to prove this concentration inequality via the technique of exchangeable pairs introduced by (Chatterjee, 2007). Consider, again, the *anti-symmetric* function, $F : S \times S \to \mathbb{R}$,

$$F(\boldsymbol{\tau}, \boldsymbol{\tau}') = \frac{1}{2}\sum_{i=1}^n (y_{ih(i)}(\boldsymbol{\tau}) + y_{ih(i)}(\boldsymbol{\tau}'))(\tau_i - \tau_i'),$$

and an assignment $\boldsymbol{\sigma}$ drawn from the Ising model truncated by $S$. We construct a new assignment $\sigma'$, via taking one-step of the Glauber dynamics over the Markov random field. The value of $F$ on $(\boldsymbol{\sigma}, \boldsymbol{\sigma}')$ simplifies as,

$$F(\boldsymbol{\sigma}, \boldsymbol{\sigma}') = z_{ih(i)}(\boldsymbol{\sigma})(\sigma_I - \sigma_I').$$

Define the function $f(\boldsymbol{\sigma})$ as the *conditional* expectation of $F(\boldsymbol{\sigma}, \boldsymbol{\sigma}')$ with respect to $\boldsymbol{\sigma}$, that is

$$\begin{aligned}
f(\boldsymbol{\sigma}) = \mathbb{E}_I\left(F(\boldsymbol{\sigma}, \boldsymbol{\sigma}')|\boldsymbol{\sigma}\right) &= \frac{1}{n}\sum_{i=1}^n y_{ih(i)}(\boldsymbol{\sigma})(\sigma_i - \mathbb{E}(\sigma_i|\boldsymbol{\sigma}_{-i})) \\
&= \frac{1}{n}\sum_{i\in\mathcal{F}(\sigma)} y_{ih(i)}(\boldsymbol{\sigma})(\sigma_i - \tanh(\beta^* m_i(\boldsymbol{\sigma})))
\end{aligned}$$

To show prove the desired result, it suffices to show a bound on the second moment of $f(\boldsymbol{\sigma})$. Observe that $(\boldsymbol{\sigma}, \boldsymbol{\sigma}')$ is indeed an exchangeable pair as

$$\mathbb{E}_{\beta^*}[f(\boldsymbol{\sigma})^2] = \mathbb{E}_{\beta^*,I}[f(\boldsymbol{\sigma})F(\boldsymbol{\sigma}, \boldsymbol{\sigma}')] = \mathbb{E}_{\beta^*,I}[f(\boldsymbol{\sigma}')F(\boldsymbol{\sigma}', \boldsymbol{\sigma})].$$

Moreover, the anti-symmetric nature of $F(\boldsymbol{\sigma}, \boldsymbol{\sigma}')$ implies $\mathbb{E}_{\beta^*,I}[f(\boldsymbol{\sigma}')F(\boldsymbol{\sigma}', \boldsymbol{\sigma})] = -\mathbb{E}_{\beta^*,I}[f(\boldsymbol{\sigma}')F(\boldsymbol{\sigma}, \boldsymbol{\sigma}')]$. These facts combine to recast $\mathbb{E}_{\beta^*}[f(\boldsymbol{\sigma})]$ as follows,

$$\begin{aligned}
\mathbb{E}_{\beta^*}\left[f(\boldsymbol{\sigma})^2\right] = \mathbb{E}_{\beta^*,I}[f(\boldsymbol{\sigma})F(\boldsymbol{\sigma}, \boldsymbol{\sigma}')] &= -\mathbb{E}_{\beta^*,I}[f(\boldsymbol{\sigma}')F(\boldsymbol{\sigma}, \boldsymbol{\sigma}')] \\
&= \frac{1}{2}\mathbb{E}_{\beta^*,I}\left[(f(\boldsymbol{\sigma}) - f(\boldsymbol{\sigma}'))F(\boldsymbol{\sigma}, \boldsymbol{\sigma}')\right]
\end{aligned}$$

If $\boldsymbol{\sigma} = \boldsymbol{\sigma}'$ then this expression is rendered trivially zero, and hence we need only analyse the case when $\sigma_I' = -\sigma_I$. If the redrawn coordinate $I$ is selected from the set of flippable indices, this probability is,

$$p_i(\boldsymbol{\sigma}) := \frac{\exp(-\sigma_i(\beta^* m_i(\boldsymbol{\sigma})))}{\exp(-\beta^* m_i(\boldsymbol{\sigma})) + \exp(\beta^* m_i(\boldsymbol{\sigma}))} = \mathbf{Pr}(\sigma_i' = -\sigma_i|\boldsymbol{\sigma}, I = i, i \in \mathcal{F}(\boldsymbol{\sigma}))$$

and when $I \notin \mathcal{F}(\boldsymbol{\sigma})$ this probability is zero. Using the definitions of $f(\boldsymbol{\sigma})$ and $F(\boldsymbol{\tau}, \boldsymbol{\tau}')$ above, this expression is simplified as follows, where $\boldsymbol{\sigma}^{(i)} = (-\sigma_i, \boldsymbol{\sigma}_{-i})$.

$$\begin{aligned}
\frac{1}{2}\mathbb{E}_I\left[(f(\boldsymbol{\sigma}) - f(\boldsymbol{\sigma}'))F(\boldsymbol{\sigma}, \boldsymbol{\sigma}')|\boldsymbol{\sigma}\right] &= \frac{1}{n}\sum_{i\in\mathcal{F}(\boldsymbol{\sigma})}(f(\boldsymbol{\sigma}) - f(\boldsymbol{\sigma}^{(i)}))F(\boldsymbol{\sigma}, \boldsymbol{\sigma}^{(i)})p_i(\boldsymbol{\sigma}) \\
&= \frac{1}{n}\sum_{i\in\mathcal{F}(\boldsymbol{\sigma})}(f(\boldsymbol{\sigma}) - f(\boldsymbol{\sigma}^{(i)}))y_{ih(i)}(\boldsymbol{\sigma})(\sigma_i - \tanh(\beta^* m_{h(i)}(\boldsymbol{\sigma}))p_i(\boldsymbol{\sigma}) \\
&:= \frac{1}{n}\sum_{i\in\mathcal{F}(\boldsymbol{\sigma})} T_{1i}T_{2i}
\end{aligned}$$

*Bound on $T_{1i}$:* We now bound each of term in the above expression, beginning with $T_{1i}$ where $i$ is flippable. The Taylor expansion of $f(\boldsymbol{\sigma}^{(i)})$ centered at $f(\boldsymbol{\sigma})$ yields,

$$|f(\boldsymbol{\sigma}^{(i)}) - f(\boldsymbol{\sigma})| \leq |\sigma_i - \sigma_i^{(i)}|\max_{w\in[-1,1]}\frac{\partial f}{\partial\sigma_i}((w, \boldsymbol{\sigma}_{-i})) = \max_{w\in[-1,1]} 2\cdot\frac{\partial f}{\partial\sigma_i}((w, \boldsymbol{\sigma}_{-i})),$$

where $w$ is point along the line with endpoints $\boldsymbol{\sigma}$ and $\boldsymbol{\sigma}^{(j)}$.

The partial derivative of $f(\boldsymbol{\sigma})$ with respect to $\sigma_i$ evaluated at a spin configuration $\boldsymbol{\tau} \in S$ is

$$\frac{\partial f}{\partial \sigma_i}(\boldsymbol{\tau}) = \frac{1}{n} \sum_{j \in \mathcal{F}(\boldsymbol{\tau})} \left( \left( \mathbf{1}_{i=j} - \frac{\beta^* A_{h(j)i}}{\cosh^2(\beta^* m_{h(j)}(\boldsymbol{\tau}))} \right) y_{jh(j)}(\boldsymbol{\tau}) + \left( \tau_j - \tanh(\beta^* m_{h(j)}(\boldsymbol{\tau})) \frac{\partial y_{jh(j)}(\boldsymbol{\tau})}{\partial \sigma_i} \right) \right)$$

The assumption $\|A\|_\infty \leq 1$ implies $|m_i(\boldsymbol{\tau})| \leq 1$ for all values of $i \in [n]$ and $\boldsymbol{\tau} \in \{-1,1\}^n$. Furthermore, $|\cosh(\cdot)| \geq 1$, yielding the following bound on the rescaled first term.

$$\left| \sum_{j \in \mathcal{F}(\boldsymbol{\tau})} \left( \mathbf{1}_{i=j} - \frac{\beta^* A_{h(j)i}}{\cosh^2(\beta^* m_{h(j)}(\boldsymbol{\tau}))} \right) y_{jh(j)}(\boldsymbol{\tau}) \right| \leq \left( \sum_{\{j \in \mathcal{F} | h(j)=i(\boldsymbol{\tau})\}} |y_{jh(j)}(\boldsymbol{\tau})| + \sum_{\{j \neq i | j \in \mathcal{F}(\boldsymbol{\tau})\}} |\beta^* A_{h(j)i} y_{jh(j)}(\boldsymbol{\tau})| \right)$$

It can be quickly seen that this value is at *most* $(2 + 2B)$. Likewise, $\frac{\partial z_{jh(j)}(\boldsymbol{\sigma})}{\partial \sigma_i} = 2 A_{h(j)i}$ implies a bound on the second term.

$$\left| \left( \tau_j - \tanh(\beta^* m_{h(j)}(\boldsymbol{\tau})) \right) \frac{\partial y_{jh(j)}(\boldsymbol{\tau})}{\partial \sigma_i} \right| \leq \left| \left( \tau_j - \tanh(\beta^* m_{h(j)}(\boldsymbol{\tau})) \right) \right| \left| \frac{\partial y_{jh(j)}(\boldsymbol{\tau})}{\partial \sigma_i} \right| \leq 4 |A_{h(j)i}|$$

Combining these two bounds yields

$$|T_{i1}| \leq 2 \max_{w \in [-1,1]} \left| \frac{\partial f}{\partial \sigma_i}((w, \boldsymbol{\sigma}_{-i})) \right|$$

$$\leq \max_{w \in [-1,1]} \frac{2}{n} \left( (2 + 2B) + \sum_{j \in \mathcal{F}(\boldsymbol{\tau})} 4 |A_{h(j)i}| \right)$$

$$\leq \frac{2}{n} \left( (2 + 2B) + 4 \right)$$

$$\leq \frac{(12 + 4B)}{n}$$

*Bound on $T_{2i}$:* Recall $|y_{ih(i)}(\boldsymbol{\sigma})| \leq 1$ for all $i \in [n]$ and $\boldsymbol{\sigma} \in \{-1,1\}^n$ and $|\tanh(x)| \leq 1, \forall x \in \mathbb{R}$. Then

$$|T_{2i}| = |y_{ih(i)}(\boldsymbol{\sigma})(\sigma_i - \tanh(\beta^* m_{h(i)}(\boldsymbol{\sigma})) p_i(\boldsymbol{\sigma})| \leq 4$$

*Putting together the pieces:* We are now ready to construct our final bound on $\mathbb{E}_{\beta^*}(f(\boldsymbol{\sigma})^2)$.

$$\mathbb{E}_{\beta^*}(f(\boldsymbol{\sigma})^2) = \frac{1}{2} \mathbb{E}_{\beta^*,I} \left( (f(\boldsymbol{\sigma}) - f(\boldsymbol{\sigma}')) F(\boldsymbol{\sigma}, \boldsymbol{\sigma}') \right)$$

$$= \frac{1}{2n} \mathbb{E}_{\beta^*} \left( \sum_{i=1}^n T_{1i} T_{2i} e_i(\boldsymbol{\sigma}) \right)$$

$$= \frac{(24 + 8B)}{n}$$

This directly implies that

$$\mathbb{E}_{\beta^*} \left[ \left( \sum_{i=1}^n m_i^2(\boldsymbol{\sigma}) e_i(\boldsymbol{\sigma}) - \sum_{i=1}^n \mathbb{E}_{\beta^*} \left[ m_i(\boldsymbol{\sigma}) e_i(\boldsymbol{\sigma}) | \boldsymbol{\sigma}_{-h(i)} \right] \right)^2 \right] \leq (24 + 8B)n$$

Applying Chebyshev's inequality to this term, yields a bound in probability that the second derivative deviates far from its conditional mean.

$$\mathbf{Pr}_{\beta^*}\left[\left(\sum_{i=1}^n m_i^2(\boldsymbol{\sigma})e_i(\boldsymbol{\sigma}) - \sum_{i=1}^n \mathbb{E}_{\beta^*}\left[m_i(\boldsymbol{\sigma})e_i(\boldsymbol{\sigma})|\boldsymbol{\sigma}_{-h(i)}\right]\right)^2 \geq n^{1.1}\right] \leq \frac{(24+8B)}{n^{0.1}}$$

$$\mathbf{Pr}_{\beta^*}\left[\left|\sum_{i=1}^n m_i^2(\boldsymbol{\sigma})e_i(\boldsymbol{\sigma}) - \sum_{i=1}^n \mathbb{E}_{\beta^*}\left[m_i(\boldsymbol{\sigma})e_i(\boldsymbol{\sigma})|\boldsymbol{\sigma}_{-h(i)}\right]\right| \geq n^{0.55}\right] \leq \frac{(24+8B)}{n^{0.1}}$$

$$\mathbf{Pr}_{\beta^*}\left[\sum_{i=1}^n m_i^2(\boldsymbol{\sigma})e_i(\boldsymbol{\sigma}) \leq \frac{n\exp(-B)(1-\delta)}{\Delta(4kd\Delta)^2} - n^{0.55}\right] \leq \frac{(24+8B)}{n^{0.1}}$$

$$\mathbf{Pr}_{\beta^*}\left[\sum_{i=1}^n m_i^2(\boldsymbol{\sigma})e_i(\boldsymbol{\sigma}) \leq \frac{2n\exp(-B)(1-\delta)}{\Delta(4kd\Delta)^2}\right] \geq 1 - o(1)$$

$$\square$$

## F    APPLICATIONS

In this brief section, we establish a connection between the notion of fatness, as introduced in the context of truncated Boolean product distributions (Fotakis et al., 2021), and the Ising measure conditioned on the solutions to a $k$-CNF formula. Specifically, we show that this truncated Ising measure satisfies the combinatorial conditions required for fatness, thereby extending the fatness framework beyond the setting of product distributions. We recound the definition of an $\alpha-$fat distribution below.

**Definition 1** ($\alpha$-fat Distributions (Fotakis et al., 2021)). *A truncated boolean distribution $D_S$ is $\alpha$-fat if for all coordinates $i \in [n]$ there exists some $\alpha > 0$ such that*

$$\mathbf{Pr}_{x \sim D_S}\left[(x_1, \ldots, x_{i-1}, -x_i, x_{i+1}, \ldots, x_n) \in S\right] \geq \frac{1}{2}.$$

**Corollary F.1.** *Given an Ising model $\mathbf{Pr}_{\beta,S}$, satisfying Assumption 1, whose measure is truncated to the solutions $S$ of a $k$-SAT formula such that*

$$k \geq \mathcal{O}\big(3\Delta^3(1 + \log(d^2k + 1))\big),$$

*then the distribution is $\frac{1}{2}$-fat, i.e.,*

$$\mathbf{Pr}_{\beta,S}\left[(-\sigma_i, \boldsymbol{\sigma}_{-i}) \in S\right] \geq \frac{1}{2}, \quad \text{for all } i \in [n].$$

*Proof.* This is a direct consequence of Lemma 3.5.                    $\square$

