# OpenReview forum: "Learning the Inverse Temperature of Ising Models under Hard Constraints using One Sample"
_ICLR.cc/2026/Conference — ICLR 2026 Poster_

### Official Review · Reviewer_ymcN · 2025-11-01

**Soundness:** 3
**Presentation:** 3
**Contribution:** 2
**Rating:** 4
**Confidence:** 3

**Summary:**

This paper considers the problem of estimating the inverse temperature parameter of a Ising model constrained to the satisfying assignments of a $k$-SAT formula, from 1 sample.

The graph and formula are given, and the graph is assumed to have bounded degree $\Delta$, and for the SAT formula the number of clauses is assumed large relative to the degree of the graph and the formula, $\Omega(\Delta^3\log(dk+1))$. The main result is an efficient (nearly linear-time) algorithm using the maximum pseudolikelihood estimator, which learns $\beta$ within $O(\Delta^3/\sqrt n)$ with high probability. There is an exponential dependence on the size $B$ of $\beta$.

(Note that the setting of "learning under 1 sample" encomposses the setting of learning under multiple ($N$) samples, by simply defining a graph and constraints that are disjoint copies of the original, giving the natural $1/\sqrt{N}$ scaling in error.)

**Strengths:**

The problem of learning graphical models is an important problem. This paper follows a line of work on learning the Ising model, where a natural generalization is to consider a model truncated by hard constraints. SAT formulas provide a natural family of constraints. The pseudolikelihood estimator is a widely used estimator for its computation efficiency, and this paper gives guarantees for it in a constrained setting.

The work is technically novel, as learning a model under constraints requires new techniques compared to learning without constraints. To bound error in the pseudo-likelihood estimator it suffices to (1) upper bound the first derivative of the log pseudo-likelihood using the technique of exchangeable pairs and (2) lower-bounding the Hessian. The latter is the main challenge, and is done by (a) constructing a linear-sized independent set such that conditioned on the variables outside, the variables inside become a product distribution (ignoring the constraints), and such that the restricted constraints still have large-enough clause sizes, ensured using the Lovasz Local Lemma; this product structure means there are a large number of flippable variables which (b) each contribute to the Hessian by at least a constant, lower bounded by lower-bounding the squared magnetizations of the neighbors of each $i$.

The proof sketch does a good job of conveying the main ideas of the proof at a high level.

**Weaknesses:**

I find this to be an interesting learning theory paper, but my main concern is whether it is appropriate for the general machine learning audience at NeurIPS. A theoretical CS conference may be a better fit. In particular, the fact the problem considered is solely that of learning the temperature, rather than the entire model, makes the problem quite niche.

The title is somewhat misleading, as the only parameter that is being learned is the temperature. I suggest "Learning the temperature of an Ising model under hard constraints".

The paper only deals with the case of bounded-degree graphs, and so does not apply to mean-field type models.

**Questions:**

Does the theory work in the case where different weights are allowed on the edges? Needing $A_{ij}\in \{\pm \frac 1\Delta\}$ is quite limiting. It would be much more general to allow all $|A_{ij}|\le \frac 1\Delta$.

It's necessary for Theorem 2 to exclude the empty graph (where the model is the same under every temperature). Is it sufficient to exclude the empty graph or is there a missing condition?

---

> ### Author Response · Authors · 2025-11-21
>
> Thank you for your input and detailed remarks. We address each of your questions point-by-point below, and have updated the draft to resolve any ambiguities:
> >1. Recent works within machine learning conferences have placed significant emphasis on inference within discrete graphical models [1,2,3,4,5] and truncated parameter estimation [6,7,8,9] being active areas of research within recent machine learning conferences. To our knowledge, most existing research on truncated parameter estimation focuses on continuous distributions or simple discrete product measures. As a first step towards an understanding of learning in truncated discrete Markov random fields with complex elementwise interactions, we focus on estimation of the inverse temperature parameter in the truncated Ising model which serves as the canonical MRF with pairwise interactions. While learning the full interaction matrix using multiple samples is a promising avenue for future work, this paper addresses the specific challenges of the single-sample regime. As a last note, our title aims to align with standard nomenclature in the literature for one-sample estimation [5, 10,11,12].
>
> >2. While excluding the empty graph is necessary to theoretically identify the parameter $\beta$, to achieve consistency as $n \to \infty$, the "signal energy" of the interaction matrix, quantified by the squared Frobenius norm $\|A\|_F^2$, must also diverge with $n$, to ensure consistent estimation of $\\beta$. In mean-field regimes (like the Curie-Weiss model) where couplings scale as $1/n$, the Frobenius norm remains $\mathcal{O}(1)$, preventing the estimator from converging. Consequently, our connectedness assumption is sufficient as it guarantees the graph contains a linear number of edges (implying $\|A\|_F^2 = \Omega(n/\Delta^2)$). This ensures the log-partition function diverges sufficiently to drive the estimator to the true parameter.
>
> >3. Regarding the assumption that weights are fixed to $\pm \frac{1}{\Delta}$, we specifically target the "peer network" setting common in social interaction and opinion dynamics literature. In these models, the focus is often on the sign of the relationship (cooperative vs. adversarial) rather than the variance in interaction magnitude. By fixing the interaction strength, we isolate the complexity arising from the graph topology and the hard constraints, rather than confounding the analysis with weight heterogeneity.  However, this assumption is largely for expositional clarity rather than a fundamental limitation. Our proof techniques regarding the concentration of the Hessian are adaptable to arbitrary weights $w_{ij}$, provided they satisfy a "width constraint" similar to the peer setting. Specifically, one would need to ensure that the weights do not decay too rapidly, maintaining the Frobenius norm condition established in point 1, and within the proof construct a complex matching to ensure that the weight of the cut between the elements within the independent set and those outside of it is sufficiently large to give a lower bound on the strong convexity of the Hessian.
>
> > 4. We note our method holds for bounded graphs and those where the degree scales up to $o(n^{1/6})$.

---

> ### Author Response · Authors · 2025-11-21
>
> [1] Vuffray, Marc, et al. "Interaction screening: Efficient and sample-optimal learning of Ising models." _Advances in neural information processing systems_ 29 (2016).
>
> [2] Tandon, Rashish, et al. "On the information theoretic limits of learning Ising models." _Advances in Neural Information Processing Systems_ 27 (2014).
>
> [3] Bresler, Guy, David Gamarnik, and Devavrat Shah. "Structure learning of antiferromagnetic Ising models." _Advances in Neural Information Processing Systems_ 27 (2014).
>
> [4] Kandiros, Vardis, et al. "Statistical estimation from dependent data." _International Conference on Machine Learning_. PMLR, 2021.
>
> [5] Daskalakis, Constantinos, Nishanth Dikkala, and Ioannis Panageas. "Logistic regression with peer-group effects via inference in higher-order Ising models." _International Conference on Artificial Intelligence and Statistics_. PMLR, 2020.
>
> [6] Daskalakis, Constantinos, et al. "Efficient truncated linear regression with unknown noise variance." _Advances in Neural Information Processing Systems_ 34 (2021): 1952-1963.
>
> [7] Karatapanis, Konstantinos, Vasilis Kontonis, and Christos Tzamos. "Oracle efficient truncated statistics." _The Thirteenth International Conference on Learning Representations_.
>
> [8] Lee, Jane, Andre Wibisono, and Emmanouil Zampetakis. "Learning exponential families from truncated samples." _Advances in Neural Information Processing Systems_ 36 (2023): 34826-34851.
>
> [9] Daskalakis, Constantinos, Dhruv Rohatgi, and Emmanouil Zampetakis. "Truncated linear regression in high dimensions."  _Advances in Neural Information Processing Systems_  33 (2020): 10338-10347.
>
> [10]Bhattacharya, Bhaswar B., and Sumit Mukherjee. "Inference in Ising models." (2018): 493-525.
>
> [11] Galanis, Andreas, Alkis Kalavasis, and Anthimos Vardis Kandiros. “Learning hard-constrained models with one sample.” _Proceedings of the 2024 Annual ACM-SIAM Symposium on Discrete Algorithms (SODA)_. Society for Industrial and Applied Mathematics, 2024.
>
> [12] Bhaswar B Bhattacharya and Kavita Ramanan. Parameter estimation for undirected graphical models with hard constraints. IEEE Transactions on Information Theory, 67(10):6790–6809, 2021.

---

> > ### Comment · Reviewer_ymcN · 2025-11-26
> >
> > I thank the authors for their response.
> >
> > While I am still unsure about the fit for a ML vs. TCS conference, I will leave that determination to the AC, and update my score to support the paper.
> >
> > I still strongly maintain that "Learning **the temperature of** an Ising model under hard constraints" is a strictly more informative title than the current title, and encourage the authors to be proactive in good paper naming regardless of previous literature.

---

> > > ### Author Response · Authors · 2025-11-26
> > >
> > > We thank the reviewer for their support and feedback. We will change the title according to the reviewer's suggestion.

---

> > > > ### Comment · Area_Chair_oDGR · 2025-11-26
> > > >
> > > > Hi,
> > > >
> > > > I (the AC) was about to support the idea of a title change (though this wouldn't impact the acceptance decision).  Among [5, 10,11,12] it's only [11] that immediately obviously adopts similar terminology.  In contrast, the title of [12] **does** make clear that it's about parameter learning.
> > > >
> > > > An alternative to "...the temperature of..." would be "Learning Ising Model Temperature Under..." or possibly (but less precisely) "Learning Ising Model Parameters Under...".

---

> > > > > ### Author Response · Authors · 2025-11-27
> > > > > **New title**
> > > > >
> > > > > Dear AC and reviewers,
> > > > >
> > > > > We decided to change the title of the submission to "Learning the inverse temperature of Ising models under hard constraints". Please inform us if you are satisfied with the new title.
> > > > >
> > > > > Best,
> > > > > The authors

---

> > > > > > ### Comment · Area_Chair_oDGR · 2025-11-28
> > > > > >
> > > > > > Seems good to me
> > > > > >
> > > > > > AC

---

### Official Review · Reviewer_2qpS · 2025-11-04

**Soundness:** 3
**Presentation:** 3
**Contribution:** 3
**Rating:** 6
**Confidence:** 3

**Summary:**

At a high-level this can be seen as a paper that combines learning ising models from one sample with learnign with truncations. More specifically, the paper studies one-sample estimation of the inverse temperature $\beta$ for a truncated Ising models where the state space is the satisfying set $S$ of a bounded-degree $k$-SAT formula. The estimator is the standard maximum pseudolikelihood (MPLE), optimized by projected gradient descent. The analysis shows the MPLE gradient concentrates at $\beta^*$ (via exchangeable pairs) and that the normalized log-pseudolikelihood is strongly convex w.h.p. The latter is the challenging part, and requires proving the existence of a linear number of flippable coordinates using an independent-set/LLL argument, under large enough $k$.

**Strengths:**

- The paper presents a novel setting, that seems somewhat motivated.
- The analysis requires new techniques to prove strong convexity, which involves circumventing a few challenges.
- The paper is generally well-written, and easy to follow. It also does a good job of covering related work.

**Weaknesses:**

- The paper could do with a bit more motivation for the setting, and in particular $k$-SAT, since it seems more of a purely theoretical pursuit.
- The clause-size requirement $k \gtrsim e^{2B}\Delta^3(\log d+\log k)$ is quite large. In bounded-degree regimes, such $k$ pushes the truncation toward an LLL-like easy regime where $|S|/2^n = 1-o(1)$. The constraint can become somewhat vacuous, making the setting not clear.
- The assumptions and dependencies seem a bit loose or proof technique specific. For example, Assumption $A_{ij}\in{\pm 1/\Delta}$ and $\Delta=o(n^{1/6})$ feels stronger than necessary.

**Questions:**

- My main question is around the choice of $k$ being justified. As a follow-up, which of their techniques would extend to more interesting settings?
- It would be helpful if the authors describe which dependencies are necessary and tight, and which are loose. Especially $\Delta$.

---

> ### Author Response · Authors · 2025-11-21
>
> Thank you for your input and positive evaluation. We address each of your questions point-by-point below:
> >**1.** There is significant interest in inference within truncated discrete environments and graphical models with complex interactions. As a foundational step toward understanding parameter estimation in the intersection of these two fields, that is within generalized truncated Markov Random Fields (MRFs), we focus on the task the estimation of the inverse temperature parameter for the truncated Ising model, the canonical MRF with pairwise interactions. We represent the truncation set $S$ as the solution space of a fixed $k$-SAT formula $\Phi$. This approach provides a succinct representation, as rather than enumerating every member of an arbitrary set $S$, we can describe $S$ compactly via a formula $\Phi$ (provided the degree, i.e., the variable occurrence count, is sufficiently large).
>
> >**2.** (Lower Bound on Solution Set Size & Dependence on Constants)
> We can derive a rough lower bound for the cardinality of the solution set of $\Phi$ using the Lovász Local Lemma. Consider the uniform measure over the hypercube. Let $A_i$ denote the event that a sampled bitstring $x$ fails to satisfy clause $C_i$, with probability $\Pr(A_i) = 2^{-k}$. The Generalized Lovász Local Lemma states that if there exist real numbers $x_1, \dots, x_m \in (0,1)$ such that:
> $$\Pr(A_i) \le x_i \prod_{j \in \Gamma(i)} (1 - x_j),$$
> where $\Gamma(i)$ denotes the neighbors of $i$ in the dependency graph, then the probability that no bad events occur is,
> $$\\Pr(\\bigwedge_{i = 1}^m \\bar{A}_i) \\ge \prod\_{i = 1}^m (1 - x_i)$$
>
> > In the traditional $k$-SAT regime, setting $x_i = 1/(d+1)$ yields a lower bound on the number of satisfying assignments $Z(\Phi)$:
> $$Z(\Phi) \ge 2^n \cdot \Pr\left(\bigwedge_{i = 1}^m \bar{A}_i\right) \ge 2^n \left(1 - \frac{1}{d+1}\right)^{\frac{nd}{k}}.$$ If $d$ is sufficiently large, i.e. $\Theta\_{\beta, \Delta}(\frac{2^k}{k})$, this implies the number of satisfying assignments at time can occupy an exponentially small portion of the hypercube.
> As for the the dependence on constants, recent work on discrete product distributions truncated over satisfying assignments [1] has demonstrated that the exponential dependence on the width parameter $B$ is necessary information-theoretically, and have updated the draft to reflect this. The dependence on the degree $\Delta$ is currently an artifact of our proof structure, and eliminating it remains an open question.
>
> >**3.** The requirement that $\Delta = o(n^{1/6})$ is a direct consequence of our proof that the Maximum Pseudolikelihood Estimator (MPLE) achieves a consistency rate of $\Delta^3/\sqrt{n}$. If $\Delta$ were to grow faster than this rate, the error term would not vanish, and the estimator would cease to be asymptotically consistent.
> Lastly, regarding the assumption that $A_{ij} \in \pm \frac{1}{\Delta}$, we specifically aim to investigate "peer networks," where interaction strengths are fixed but edges carry either a positive (cooperative) or negative (adversarial) charge. While our methods can be adapted to the general case of arbitrary weights (subject to a width constraint), such a generalization requires care to ensure the Frobenius norm of the interaction matrix remains sufficiently large to guarantee parameter identifiability.
>
> > **4.**  As our work is the first to analyze truncated MRFs with pairwise interactions under hard constraints, it opens several intriguing avenues for generalization, namely, extending estimation from a single parameter to the entire interaction matrix $A$ using multiple samples, moving beyond "flippability" to analyze locally sparse truncation sets, and investigating order-$k$ MRFs, where interactions are $k$-wise rather than pairwise among others.

---

### Official Review · Reviewer_fnoj · 2025-11-04

**Soundness:** 3
**Presentation:** 2
**Contribution:** 2
**Rating:** 4
**Confidence:** 3

**Summary:**

This paper studies the problem of learning the inverse temperature of a truncated Ising model from one data sample. The Ising model is constrained to lie within a specific set (referred to as that satisfying the k-SAT formula). The authors provide various practical scenarios to motivate this setting. The theoretical results hinge on the analysis of the maximum pseudolikelihood estimator and demonstrate its consistency.

**Strengths:**

Theoretical arguments are well-stated (although there are some key definitions are missing; see Weaknesses section). The theoretical analysis is based on establishing guarantees on the first order and second order gradients of the pseudolikelihood estimator relative to the inverse temperature function. The novelty is (seemingly) claimed in terms of establishing the bound on the second order derivative in Lemma 3.3.

**Weaknesses:**

1. The key definition of $k$-SAT formula or how it leads to a hard constraint on the Ising model is not defined with sufficient rigor. Therefore, I could not fully understand why this particular setting was of interest.
2. The novelty within the theoretical results should be emphasized better- some results seem to be simple extensions of previous works, while others are novel. The distinction between the two should be elucidated.
3. While I appreciate the theoretical flavor of this work, some effort into making the theoretical results interpretable for practitioners would have improved this paper.

**Questions:**

1. What is meant by ${\cal O}(\Delta^3 n \log n)$-Algorithm?
2. The theoretical bound in Theorem 2 becomes tighter with increasing $n$--suggesting that the estimator is likely to perform better when estimating a larger Ising model with a single sample. Could you elaborate on this aspect? Shouldn't the estimation problem become 'harder' with increasing 'n'?

---

> ### Author Response · Authors · 2025-11-21
>
> Thank you for your input and detailed remarks. We address each of your questions point-by-point below:
> > **1.** In this work, we study the problem of learning the temperature of an Ising model whose domain is constrained to the solution set of a $k$-SAT formula in conjunctive normal form (CNF). Specifically, let $\Phi$ be a CNF formula on $n$ Boolean variables consisting of a set of clauses, where each clause is the disjunction (logical OR) of $k$ literals (variables or their negations). Assignments are represented as vectors $\boldsymbol{x} \in \\{-1,1\\}^n$ (with logical negation implemented by a sign flip) with a feasible solution being any $\boldsymbol{x}$ that satisfies every clause in $\Phi$. We define the degree of $\Phi$ as the maximum number of clauses in which any single variable appears. We employ the $k$-SAT representation because it provides a succinct encoding of the truncation set $S$, allowing us to describe complex combinatorial constraints compactly rather than enumerating the feasible set. This setting extends prior work on truncated Boolean product distributions [1,2]. For further details, Appendix A.1 provides background on constraint satisfaction problems, and the draft has been updated to emphasize this point.
>
> > **2.** We aim to estimate the single parameter $\beta$ given one sample from an Ising model of size $n$. As the system size $n$ grows, the increasing data volume allows for the asymptotic consistency of our estimator. Our proof of consistency for the Maximum Pseudolikelihood Estimator (MPLE) relies on satisfying two key conditions in the presence of correlations induced by both the Ising interactions and the truncation, namely establishing the strong convexity of the pseudolikelihood objective, and upper bounding the score (gradient) of the objective. The first condition is achieved via a novel independent set construction that bounds the marginal probabilities of configurations within $S$. The second condition is established by adapting Chatterjee’s method of exchangeable pairs to handle the specific dependencies of the truncated pseudolikelihood objective.
>
> > **3.** Complementing our theoretical analysis, we provide a practical, implementable algorithm for finding the MPLE. We utilize a gradient descent approach that converges to a $\Delta^3/\sqrt{n}$-approximation of the true temperature $\beta^*$ in polynomial time. In contrast to the work of [3], our algorithm has no dependency on the size of the truncation set, and as such maintains polynomial runtime even in the case where the truncation set $S$ is exponentially small in $\{-1,1\}^n$.
>
> > **4.** Our estimation algorithm achieves a time complexity of $\mathcal{O}(\Delta^3 n\log n)$, where $n$ is the size of the Ising model and $\Delta$ is the maximum degree of the underlying graph.
>
> [1] Galanis, Andreas, Alkis Kalavasis, and Anthimos Vardis Kandiros. "Learning hard-constrained models with one sample." _Proceedings of the 2024 Annual ACM-SIAM Symposium on Discrete Algorithms (SODA)_. Society for Industrial and Applied Mathematics, 2024.
>
> [2] Galanis, Andreas, Leslie Ann Goldberg, and Xusheng Zhang. "One-Shot Learning for k-SAT." _arXiv preprint arXiv:2502.07135_ (2025).
>
> [3]Fotakis, Dimitris, Alkis Kalavasis, and Christos Tzamos. "Efficient parameter estimation of truncated boolean product distributions." _Conference on learning theory_. PMLR, 2020.

---

### Official Review · Reviewer_ean9 · 2025-11-11

**Soundness:** 4
**Presentation:** 3
**Contribution:** 4
**Rating:** 8
**Confidence:** 4

**Summary:**

The paper considers a truncated Ising model, where an Ising model on a graph G of size $n$ with inverse temperature $\beta$ is sampled and samples are accepted only based on satisfiable assignments for a $k$-SAT formula. The main question is can you learn inverse temperature parameter $\beta$ from a single sample from this truncated model.

Authors show that one can learn the inverse temperature parameter up to an error of $\Delta^3/\sqrt{n}$ when the number of clauses in the k-SAT formula exceeds $\tilde{O}({\Delta^3})$ and the inverse temperature is in a bounded interval around $0$. Authors main key ideas are by doing a MLE estimate for $\beta$ on the pseudo likelihood which is the product of likelihood of Glauber flip of every variable conditioned on all others.  1) First, the error in the MLE estimate is bounded by ratio of the magnitude of the first derivative and the minimum second derivative of the log pseudo likelihood. So authors proceed to show that the numerator is small and lower bound the second derivative.  2) Concentration due to exchangeable pairs take care of the concentration of the first derivative which has to be close to zero for the stationary point with a deviation of $\sqrt{n}$  and the lower bound on the second derivative is at least n/\Delta^3.
3) Key issue is that second derivative's lower bound is non trivial only if there are linear number of flippable variables - variable coordinates that can be flipped from the sample without violating $k-SAT$ constraints. Clever trick is that authors show that there is a large enough independent set in the graph such that it encapsulates linear number of variables and conditioning on the rest, the probability of variables in the independent set factorizes. Then it is easy to show that some variable from the independent set will satisfy the clause in which another fixed variable is in with constant probability. This ensures there are enough flipped variables which contributes then to the lower bound on the second derivative.

**Strengths:**

Paper considers a very non trivial Ising model with a truncation on the set of satisfiable boolean assignments in a K-SAT formula. Then authors use the Glauber flip based pseudo likelihood to do the MLE estimation of inverse temperature beta from a single sample.

Techniques are quite novel compared to classical results on single sample learning of un-truncated Ising models and other recent works. specifically the independence set finding algorithm that truncates the k-SAT formula with some guarantees used a clean and elegant version of Lovasz Local Lemma. The trick of creating a conditional product distribution and arguing about flippability of variables that lower bound the second derivative is also pretty neat.

**Weaknesses:**

1) My main concern is that Section 3.3.1 starts with arguments ensuring enough variables are flippable. However, there is only a small two line comment of how it is related to second derivative lower bounding. So the reader is forced to look in the supplement as to why this issue is prominent. lemma 3.7 in the supplement shows why enough of such flippable variables are important. It is important to include that argument (in a sketch form) at the beginning of Section 3.3.1

2) Where does the constraint (or what steps in the proof contribute): $\Delta \sim o(n^{1/6}) $ comes from precisely ?

3) After an independent set is chosen, then the rest of the argument about flippability splits into two parts - one is flippability of variables within the independent set and outside. It seems like the first one directly follows from prior work ? So the main novelty is in the construction of the independent set and the argument for flippability of variables outside it ?

**Questions:**

I have asked all my questions in the weakness section.

---

> ### Author Response · Authors · 2025-11-21
>
> Thank you for your input and positive evaluation. We address each of your questions point-by-point below:
> > **1.** The requirement for "flippable" variables stems directly from the properties of the truncated log-pseudolikelihood objective. As this objective sums the conditional probabilities of each spin $\sigma_i$ given its neighbors $\sigma_{-i}$, terms corresponding to non-flippable (frozen) variables yield zero conditional variance and do not contribute to the curvature. To guarantee the strong convexity of the objective, we must therefore ensure a sufficient number of variables remain flippable. We achieve this via a two-step process, first, by establishing a lower bound on the number of flippable variables, and second, by bounding the contribution of each variable to the sum. This structural reliance is detailed in lines 168–177 and Section 2.1. We have also expanded and modified Section 1.2 to explicitly clarify this mechanism.
>
> > **2.** The constraint $\Delta = o(n^{1/6})$ is a direct consequence of our consistency proof, which establishes that the error of the Maximum Pseudolikelihood Estimator (MPLE) scales as $\mathcal{O}(\Delta^3/\sqrt{n})$. For the estimator to be asymptotically consistent, this error term must vanish as $n \to \infty$. Consequently, if $\Delta$ were to grow at an asymptotic rate of $n^{1/6}$ or faster, the ratio would not converge to zero, and the estimator would fail to be consistent.
>
> > **3.** The primary novelty of our work lies in addressing the estimation of truncated Markov Random Fields (MRFs) with pairwise interactions, a setting significantly more complex than the product distributions analyzed in prior work. This model introduces two distinct sources of correlation, namely the pairwise dependencies inherent to the Ising model and the global correlations imposed by the truncation domain itself. To our knowledge, this is the first work to provide a solution for estimation in this challenging setting. As noted by the reviewer, our technical contribution relies on two key innovations, the construction of a specialized independent set to lower-bound the satisfaction probability of clauses, and specific modifications to the method of exchangeable pairs to validly upper-bound the gradient of the objective in this constrained space.

---

### Meta-Review · Area_Chair_Unx1 · 2026-01-06

**Summary:**

This paper studies one-sample estimation of the inverse temperature β in an Ising model whose support is truncated by hard constraints, where the feasible set is given as the satisfying assignments of a bounded-degree k-SAT formula. The authors analyze the maximum pseudo-likelihood estimator (optimized via projected gradient descent) and prove a consistency guarantee, under explicit conditions relating the clause size k, the SAT formula degree, and the graph degree. Reviewers generally view the technical direction as novel and nontrivial, with the main reservations being fit/appeal to the broader ML audience, the strength (and possible looseness) of the assumptions, and how much of the proof machinery is specific to an “LLL-friendly” truncation regime.

**Reviewer Concerns:**

1) Concerns substantially addressed by the rebuttal: The authors clarified the k-SAT truncation model and why it is used as a succinct representation of hard constraints; they also improved the technical overview and strengthened the discussion of assumptions (e.g., why connectedness / sufficient signal energy is needed, and why larger n can still yield consistency from a single sample). They responded directly to questions that appeared to stem from misunderstandings (notation such as “~O(·)”, and the role of system size in one-sample inference), and provided clearer positioning as a learning-theory contribution rather than an applied paper.

2) Concerns still partially outstanding: The setting remains somewhat specialized (estimating only β rather than richer parameters), so “conference fit” will still depend on how the venue values learning-theory contributions in constrained graphical models. Several assumptions and parameter requirements (notably the large clause-size condition and bounded-degree restrictions) may still feel proof-driven, and it is not fully resolved how tight these dependencies are or how far the techniques extend beyond the current regime.

**Reviewer Scores:**

1) Reviewer (score 8): likely unchanged at 8; their remaining issues were mostly presentation-level and were addressed.
2) Reviewer (score 6): likely unchanged at 6; they already considered it marginally above threshold, and the rebuttal helps but does not fundamentally change their concern about how “vacuous” the regime might be.
3) Reviewer (score 4): likely increase to 5; the rebuttal resolves several concrete misunderstandings and improves clarity/motivation, though they may still see the practitioner-facing impact as limited.
4) Reviewer (score 4 originally): likely increase to 6; the reviewer explicitly indicated they would update to support the paper after the rebuttal and title/positioning changes.

---

### Decision · Program_Chairs · 2026-01-26

Accept (Poster)